# VERIFIED: A Video Corpus Moment Retrieval Benchmark for Fine-Grained Video Understanding

**Houlun Chen**[1], **Xin Wang**[1,2*], **Hong Chen**[1], **Zeyang Zhang**[1]
**Wei Feng**[1], **Bin Huang**[1], **Jia Jia**[1,2*], **Wenwu Zhu**[1,2*]
[1] Department of Computer Science and Technology, Tsinghua University, Beijing, China
[2] BNRIST, Tsinghua University, Beijing, China
{chenhl23,h-chen20,zy-zhang20,fw22,huangb23}@mails.tsinghua.edu.cn
{xin_wang,jjia,wwzhu}@tsinghua.edu.cn
https://verified-neurips.github.io

## Abstract

Existing Video Corpus Moment Retrieval (VCMR) is limited to coarse-grained understanding, which hinders precise video moment localization when given fine-grained queries. In this paper, we propose a more challenging fine-grained VCMR benchmark requiring methods to localize the best-matched moment from the corpus with other partially matched candidates. To improve the dataset construction efficiency and guarantee high-quality data annotations, we propose VERIFIED, an automatic VidEo-text annotation pipeline to generate captions with RelIable FInE-grained statics and Dynamics. Specifically, we resort to large language models (LLM) and large multimodal models (LMM) with our proposed Statics and Dynamics Enhanced Captioning modules to generate diverse fine-grained captions for each video. To filter out the inaccurate annotations caused by the LLM hallucination, we propose a Fine-Granularity Aware Noise Evaluator where we fine-tune a video foundation model with disturbed hard-negatives augmented contrastive and matching losses. With VERIFIED, we construct a more challenging fine-grained VCMR benchmark containing Charades-FIG, DiDeMo-FIG, and ActivityNet-FIG which demonstrate a high level of annotation quality. We evaluate several state-of-the-art VCMR models on the proposed dataset, revealing that there is still significant scope for fine-grained video understanding in VCMR. Code and Datasets are in https://github.com/hlchen23/VERIFIED.

## 1   Introduction

Video Corpus Moment Retrieval (VCMR) [1] aims to retrieve a video moment from a large untrimmed video corpus given a text query. It requires handling two subtasks: Video Retrieval (VR) [2] from a corpus and Single Video Moment Retreival (SVMR) [3, 4] within a video, which involves grasping multi-level semantic granularities across video-text and moment-text alignment. However, as shown in Figure 1(a), in the previous VCMR setting, the queries are usually coarse-grained and thus struggle to localize a video moment discriminatively, where there exists potentially relevant positive pairs [5–7] besides the ground truth, which hinders cross-modal retrieval and makes it hard for models to learn distinctive video features.

To address the problem, we propose a more challenging VCMR scenario in this paper. As shown in Fig 1(b), a fine-grained distinctive query is provided to retrieve the best-matched moment, requiring models to precisely understand the details in text descriptions and distinguish the target moments from

---

*Corresponding authors.

38th Conference on Neural Information Processing Systems (NeurIPS 2024) Track on Datasets and Benchmarks.

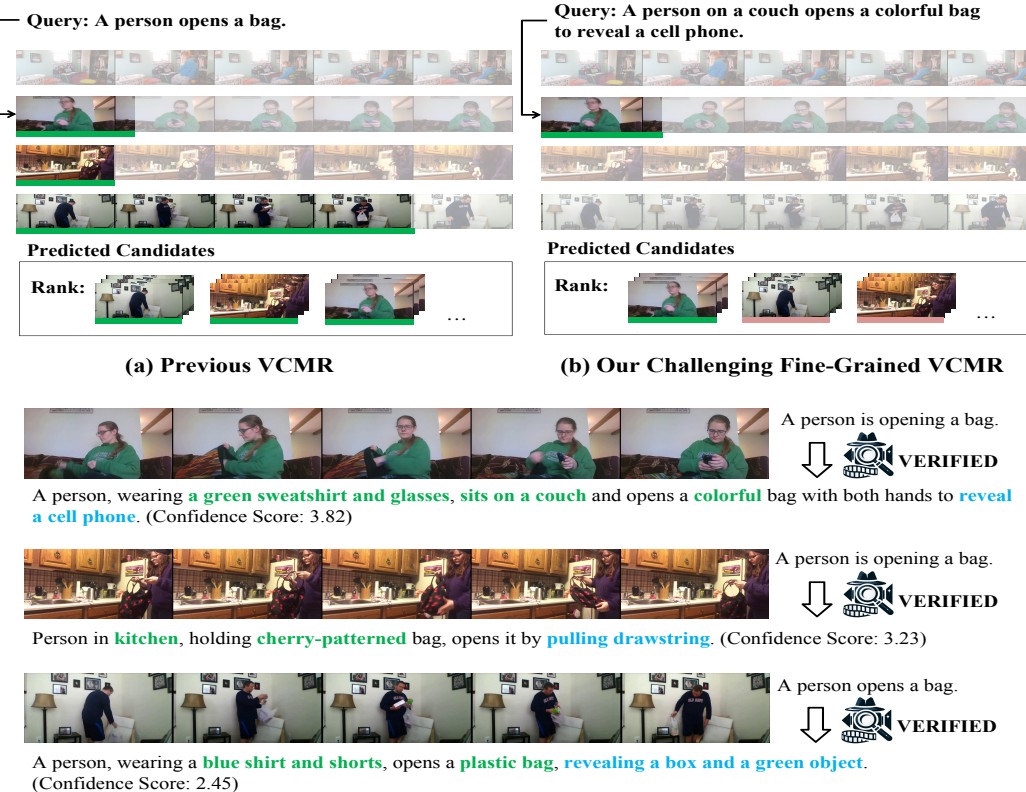

Figure 1: (a) Previous VCMR, where a query may be coarse and there are many potential positive moments (**green**) that are not annotated, making the ground truth annotations unreasonable. (b) Our Challenging Fine-Grained VCMR, where a more fine-grained query is given and the method needs to retrieve the best matched one from partially matched candidates (**pink**). (c) Our VERIFIED pipeline generates fine-grained annotations with reliable static (**green**) and dynamic (**blue**) details.

partially matched candidates. However, annotating such fine-grained video-text datasets [8–11] relies on intensive manual work and domain knowledge, limiting its productivity and scalability. Therefore, we resort to the power of recent large language model (LLM) and large multimodal model (LMM) [12–16] for automatic detailed video-clip annotation. Simply relying on the LLMs/LMMs for annotation faces the following two challenges: 1) how to extract as much fine-grained information from videos as possible remains unexplored, especially dynamic video details; 2) LLM or LMM are known to struggle with the hallucination problem, how to avoid the impact of the generated inaccurate content is also challenging.

To tackle these challenges, we propose **VERIFIED**, an automatic VidEo-text annotation pipeline to generate captions with RelIable FInE-grained statics and Dynamics. To fully utilize fine-grained visual content, we design the Statics and Dynamics Enhanced Captioning modules. Specifically, for statics, we extract foreground and background attributes with image LMM and form several statics enhanced caption candidates via LLM rewriting. For dynamics, we propose a VQA-guided dynamic detail discovering method, which guides the video LMM to focus more on dynamic changes in the video, before having LLM rewrite dynamics enhanced captions. To alleviate the impact of the inaccurate annotations caused by LLM/LMM hallucinations, we propose a Fine-Granularity Aware Noise Evaluator where we fine-tune a video foundation model [17] with disturbed hard-negative data through contrastive and matching losses, so that it can better discriminate the unreasonable annotations. We apply it to evaluate each generated video-text pair, which helps to filter out inaccurate annotations.

We construct our benchmark based on the widely adopted VCMR datasets with our VERIFIED pipeline, including Charades-STA [3], DiDeMo [4], and ActivityNet Captions [18]. As shown in

Fig 1(c), we obtain fine-grained Charades-FIG, DiDeMo-FIG, and ActivityNet-FIG to better support fine-grained VCMR, which demonstrate a high level of annotation quality. Compared to previous ones, our benchmark significantly reduces the many-to-many situations, offering more precise ground truth annotations. We evaluate several state-of-the-art VCMR models on our benchmark, and the results show that models trained on previous datasets show poor performance on the fine-grained VCMR task, while our proposed training dataset significantly improves its performance. We believe this benchmark will inspire a lot of future work for fine-grained video understanding in VCMR.

Our contributions can be summarized as follows:

1. We first define a more challenging fine-grained VCMR setting, which requires models to understand video fine-grained information precisely and learn distinctive video features.

2. We propose an automatic fine-grained video clip annotation pipeline, VERIFIED, aided by LLMs/LMMs, which fully captions fine-grained statics and dynamics in visual content, demonstrating high annotation quality.

3. We evaluate several state-of-the-art VCMR models on our benchmarks to analyze their ability to localize fine-grained queries among large video corpus, indicating several important challenges and future directions.

## 2    Related Works

**Video Annotation through Multimodal Models**. Most video-text datasets heavily rely on manual work and domain knowledge, especially for fine-grained details [8–11], limiting their scalability, particularly in video moment datasets [3, 18, 4]. Others construct large-scale datasets via web crawling [19] or ASR [20], but suffer from noisy cross-modal alignment. With the rapid advancement of multimodal foundation models and LLM, automatically annotating large-scale video-text datasets is becoming feasible [21]. InternVid [22] integrates image captioning models to caption video clips at multiple scales, while Panda-70M [23] uses multimodal teacher models to caption 70M text-annotated videos. MVid [24] automatically captions visual, audio, and speech with LLM refinement. However, they often lack fine-grained annotations, especially for dynamic details such as motions and interactions, or rely mainly on subtitles or auxiliary text labels [25]. To address this, we propose VERIFIED to automatically capture fine-grained static and dynamic details from the vision modality with quality management.

**Video Corpus Moment Retrieval**. Video moment retrieval (VMR) [3, 4, 26–37] requires localizing a matched moment within an untrimmed video for a text query and video corpus moment retrieval (VCMR) [1] extends VMR to search the target moment from a large untrimmed video corpus, requiring appropriate integration of video retrieval and moment localization. Among VCMR methods, XML [25] introduces a convolutional start-end detector with its late fusion design. CONQUER [38] integrates query context into video representation learning for enhanced moment retrieval by two stages. ReLoCLNet [39] leverages video and frame contrastive learning through separate encoder alignment. As video corpora expand, many video moments share similar semantics with subtle differences in fine-grained statics and dynamics. While some VCMR works [5–7] have explored relevance within non-ground-truth moments and texts by pseudo-positive labels or relevance-based margin, they fail to learn distinctive differences among semantically analogous clips. To address this, we propose a fine-grained VCMR scenario that requires localizing the best-matched moment among partially matched candidates for a fine-grained text query and we introduce new datasets to benchmark state-of-the-art VCMR methods.

## 3    Dataset Construction Methodology: VERIFIED

In this section, we introduce our VERIFIED pipeline to annotate fine-grained captions for video moments with reliable static and dynamic details. The annotations in the previous video moment datasets are in the form of $(V, t_s, t_e, q)$ for a moment $V[t_s : t_e]$ (designated as $v$) from $t_s$ to $t_e$ seconds in the video $V$, where $q$ is a moment-level text caption for moment $v$. In this paper, our VERIFIED pipeline constructs novel fine-grained video moment datasets in the form of $(V, t_s, t_e, \{q_i^{(s)}\}_{i=1}^{N_s}, \{c_i^{(s)}\}_{i=1}^{N_s}, \{q_i^{(d)}\}_{i=1}^{N_d}, \{c_i^{(d)}\}_{i=1}^{N_d})$. We annotate the same video moment in the previous dataset with multiple diverse captions containing rich static and dynamic fine-grained

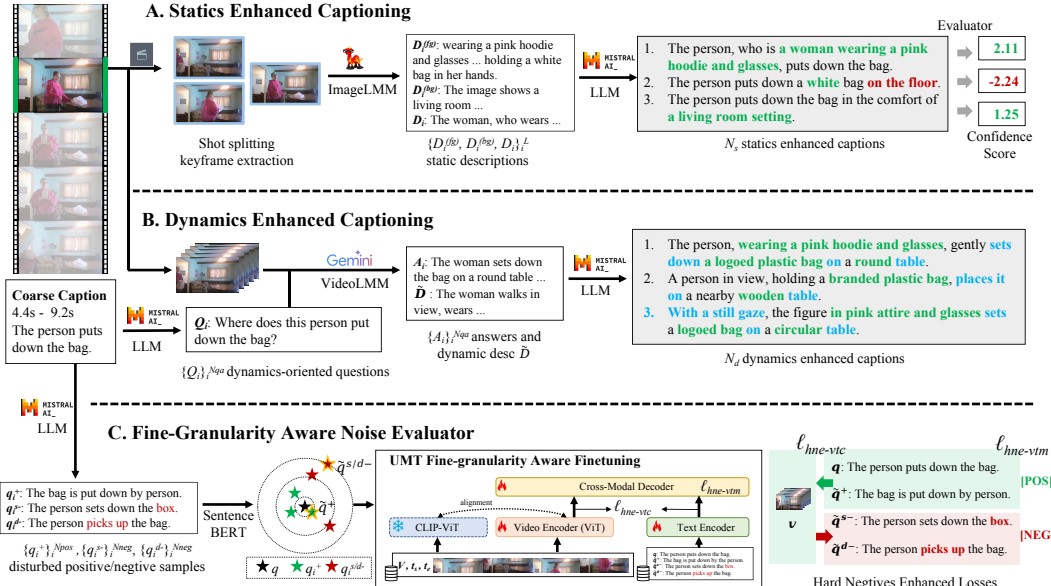

Figure 2: Our VERIFIED annotation pipeline includes two independent modules: Statics Enhanced Captioning (A) and Dynamics Enhanced Captioning (B), which generate multiple fine-grained caption candidates with static and dynamic details. Additionally, we develop a Fine-Granularity Aware Noise Evaluator (C) that generates and selects the best disturbed positive and negative samples to fine-tune UMT using hard-negative augmented contrastive and matching losses. This evaluator grades captions, assigning low confidence scores to inaccurate ones.

information. $\{q_i^{(s)}\}_{i=1}^{N_s}$ and $\{q_i^{(d)}\}_{i=1}^{N_d}$ are for fine-grained static and dynamic captions, respectively, with confidence scores $\{c_i^{(s)}\}_{i=1}^{N_s}$ and $\{c_i^{(d)}\}_{i=1}^{N_d}$. Captions within $\{q_i^{(s)}\}_{i=1}^{N_s}$ or $\{q_i^{(d)}\}_{i=1}^{N_d}$ share nearly identical coarse semantics yet they may exhibit distinct fine-grained static or dynamic details.

## 3.1 Statics Enhanced Captioning

Given the moment $(t_s, t_e)$ in the video $V$ with its original coarse caption annotation $q$, we first extract key frames from the moment $(t_s, t_e)$. Concretely, we adaptively adjust the threshold in PySceneDetect[2] to split the moment up to $L$ segments and we select the mid-time frames of these segments as key frames $\{f_i\}_{i=1}^{L}$. For each keyframe $f_i$, we prompt a strong image LMM with the inputs of the previous coarse caption $q$ as guidance and the key frame $f_i$ to describe fine-grained details of the foreground $\mathcal{D}_i^{(fg)}$ and background $\mathcal{D}_i^{(bg)}$ before generating a complete fine-grained description $\mathcal{D}_i$ of this frame.

$$\mathcal{D}_i^{(fg)}, \mathcal{D}_i^{(bg)}, \mathcal{D}_i = \text{ImageLMM}(f_i, q) \tag{1}$$

Afterward, we prompt an LLM to extract important static attributes to rephrase it to $N_s$ diverse fine-grained caption candidates $\{q_i^{(s)}\}_{i=1}^{N_s}$ as follows,

$$\{q_i^{(s)}\}_{i=1}^{N_s} = \text{LLM}(\{\mathcal{D}_i^{(fg)}, \mathcal{D}_i^{(bg)}, \mathcal{D}_i\}_{i=1}^{L}, q) \tag{2}$$

These new captions now contain rich static visual details about the video moment.

## 3.2 Dynamics Enhanced Captioning

Since it's hard for even existing strong video captioning models to capture rich enough dynamic information, we introduce video question answering (VQA) to enhance the dynamic information extraction process. We prompt an LLM to generate $N_{qa}$ relevant dynamics-oriented questions

---

[2]https://www.scenedetect.com/

$\{\mathcal{Q}_i\}_{i=1}^{N_{qa}}$ on the video moment according to the previous coarse caption.

$$\{\mathcal{Q}_i\}_{i=1}^{N_{qa}} = \text{LLM}(q) \tag{3}$$

Afterward, we feed a strong video LMM with sequential video frames and such questions to have the video LMM answer these questions $\{\mathcal{A}_i\}_{i=1}^{N_{qa}}$ before generating a complete fine-grained description $\tilde{\mathcal{D}}$ of the dynamics of the video moment.

$$\{\mathcal{A}_i\}_{i=1}^{N_{qa}}, \tilde{\mathcal{D}} = \text{VideoLMM}(\{\mathcal{Q}_i\}_{i=1}^{N_{qa}}, v, q) \tag{4}$$

Finally, we prompt an LLM to extract important dynamic details to rephrase it to $N_d$ fine-grained caption candidates $\{q_i^{(d)}\}_{i=1}^{N_d}$.

$$\{q_i^{(d)}\}_{i=1}^{N_d} = \text{LLM}(\{\mathcal{Q}_i\}_{i=1}^{N_{qa}}, \{\mathcal{A}_i\}_{i=1}^{N_{qa}}, \tilde{\mathcal{D}}, q) \tag{5}$$

### 3.3 Fine-Granularity Aware Noise Evaluator

Specifically, for a piece of original sample $(V, t_s, t_e, q)$, we prompt LLM to generate $N_{pos}$ positively rewritten captions $\{q_i^+\}_{i=1}^{N_{pos}}$, $N_{neg}$ statics disturbed negative captions $\{q_i^{s-}\}_{i=1}^{N_{neg}}$, and $N_{neg}$ dynamics disturbed negative captions $\{q_i^{d-}\}_{i=1}^{N_{neg}}$ for the previous coarse caption $q$, respectively.

$$\{q_i^+\}_{i=1}^{N_{pos}}, \{q_i^{s-}\}_{i=1}^{N_{neg}}, \{q_i^{d-}\}_{i=1}^{N_{neg}} = \text{LLM}(q) \tag{6}$$

where we prompt LLM to generate rewritten captions $\{q_i^+\}_{i=1}^{N_{pos}}$ that share the same meanings as the previous coarse caption $q$, $\{q_i^{s-}\}_{i=1}^{N_{neg}}$ that have an significant difference in some static attributes from $q$, and $\{q_i^{d-}\}_{i=1}^{N_{neg}}$ that have an significant difference in some dynamic content from $q$.

Since LLM sometimes fails to generate appropriate rewritten captions, *e.g.* some $q_i^{d-}$ might share the same meaning as $q$, we select the best one from the candidates. We adopt SentenceBERT [40] as a semantic distance measure on captions to discover the most positive caption $\tilde{q}^+$ and most negative static or dynamic one $\tilde{q}^{s/d-}$, where

$$\tilde{q}^+ = \arg\min\{\text{SentenceBERT}(\{q_i^+\}_{i=1}^{N_{pos}}, q)\} \tag{7}$$

$$\tilde{q}^{s/d-} = \arg\max\{\text{SentenceBERT}(\{q_i^{s/d-}\}_{i=1}^{N_{neg}}, q)\} \tag{8}$$

Afterward, we finetune UMT [17] in the video-text retrieval task with the hard-negatives augmented contrastive loss $\ell_c$ and matching loss $\ell_m$. The contrastive loss $\ell_c$ is

$$\ell_c = -\frac{1}{2B}\left(\sum_{i=1}^{B} \log \frac{\exp\left(s(v_i, q_i)/\tau\right)}{\sum_{j=1}^{B}\sum_{\hat{q}\in\{q_j, \tilde{q}_j^{s-}, \tilde{q}_j^{d-}\}} \exp\left(s(v_i, \hat{q})/\tau\right)} + \sum_{i=1}^{B} \log \frac{\exp\left(s(q_i, v_i)/\tau\right)}{\sum_{j=1}^{B} \exp\left(s(q_i, v_j)/\tau\right)}\right) \tag{9}$$

where $B$ is the batch size and $s$ is a similarity measure. The $\tilde{q}_i^{s/d-}$ is a hard negative sample for $v_i$ and other $\tilde{q}_j^{s/d-}$ ($j \neq i$) can be seen as trivial negative samples for $v_i$. The matching loss $\ell_m$ is

$$\ell_m = -\frac{1}{B}\sum_{i=1}^{B}\left[\log\sigma(c(v_i, q_i)) + \sum_{\hat{q}\in\mathcal{N}(v_i)}\log\left(1 - \sigma(c(v_i, \hat{q}))\right) + \sum_{\hat{v}\in\mathcal{N}(q_i)}\log\left(1 - \sigma(c(\hat{v}, q_i))\right)\right] \tag{10}$$

where $c$ is a classifier, $\sigma$ is the Sigmoid function, and $\mathcal{N}(v_i), \mathcal{N}(q_i)$ contains the negative samples for $v_i$ and $q_i$. $\mathcal{N}(v_i)$ contains sampled trivial negative sample $q_j (j \neq i)$ and the augmented hard-negatives $\tilde{q}_i^{s-}$ and $\tilde{q}_i^{d-}$ while $\mathcal{N}(q_i)$ contains only sampled trivial negative sample $v_j (j \neq i)$.

To alleviate potential harmful biases in LLM-generated texts and avoid degenerating into distinguishing the human-written text from other LLM-generated ones, we replace the previous coarse caption $q$ with the selected positively rewritten caption $\tilde{q}^+$ for fine-tuning as well with loss $\ell_c^+$ and $\ell_m^+$.

Finally, we combine all these items to derive the total loss function $\ell$.

$$\ell = \frac{\lambda_c}{2}(\ell_c + \ell_c^+) + \frac{\lambda_m}{2}(\ell_m + \ell_m^+) \tag{11}$$

After fine-tuning, we evaluate the matching confidence scores between the video moment $v$ and our generated fine-grained captions $\{q_i^{(s)}\}_{i=1}^{N_s}$ and $\{q_i^{(d)}\}_{i=1}^{N_d}$ as $\{c_i^{(s)}\}_{i=1}^{N_s}$ and $\{c_i^{(d)}\}_{i=1}^{N_d}$, with our evaluator. Captions with relatively lower scores are more likely to have inaccurate content.

## 3.4 Implementation Details

For the models used in our VERIFIED pipeline, we use Mistral-7B-Instruct-v0.2 [13] as our LLM, LLaVA-1.6-Mistral-7B [14] as our image LMM, Gemini-1.5-Pro [15] as our video LMM, all-MiniLM-L6-v2 [40] for SentenceBERT, and ViT-L/16_25M [17] version for pretrained UMT. Our VERIFIED automatically annotate video moments from DiDeMo [4], Charades-STA [3], and ActivityNet Captions [18], named DiDeMo-FIG, Charades-FIG, and ActivityNet-FIG respectively. In the Statics Enhanced Captioning, we select up to $L = 1$, $L = 1$ and $L = 5$ key frames for DiDeMo-FIG, Charades-FIG and ActivityNet-FIG. In the Dynamics Enhanced Captioning, we extract frames at 8fps for DiDeMo-FIG and Charades-FIG and uniformly sample 64 frames for ActivityNet-FIG for the video input of the video LMM, and we set the VQA pair numbers $N_{qa} = 5$. We generate $N_s = N_d = 3$ fine-grained statics or dynamics enhanced caption candidates for both modules. In the Fine-granularity Aware Noise Evaluator, we generate $N_{pos} = N_{neg} = 3$ positive/negative disturbed captions for each previous caption $q$. During fine-tuning, we sample 20 frames for each video moment, and the temperature $\tau$ is 0.07 and $\lambda_c$, $\lambda_m$ is 1. We collect around 125K disturbed samples from all previous datasets for fine-tuning with a learning rate of 1e-5 and $B = 16$ batch size for 10 epochs in a 4 A100-80G machine. After all, we grade each video-text pair with our evaluator and choose the fine-grained enhanced caption with the highest score for each video moment. We follow the previous dataset splits. The prompts used for LLM/LMM are attached to the supplementary materials.

## 3.5 Statistical Analysis and User Study

We present the statistics of our annotated datasets compared to the previous coarse ones in Tab 1, where our annotations feature a richer vocabulary and approximately twice the number of content words with various parts of speech, particularly adjectives, indicating that our fine-grained captions provide more detailed descriptions.

Table 1: Statistics of our annotated fine-grained datasets (FIG) compared to the previous ones (COG).

|  | # Vocab | | # Word | | # Noun | | # Verb | | # Adj | |
| --- | --- | --- | --- | --- | --- | --- | --- | --- | --- | --- |
|  | COG | FIG | COG | FIG | COG | FIG | COG | FIG | COG | FIG |
| Charades-FIG | 997 | 2590 | 6.21 | 15.38 | 2.33 | 4.91 | 1.18 | 2.20 | 0.04 | 1.17 |
| DiDeMo-FIG | 5586 | 8595 | 7.50 | 16.08 | 2.54 | 5.22 | 1.11 | 2.14 | 0.57 | 1.60 |
| ActivityNet-FIG | 10203 | 14769 | 13.17 | 26.19 | 3.67 | 8.68 | 2.21 | 3.16 | 0.60 | 3.01 |

* COG and FIG are short of "coarse-grained" and "fine-grained". # Vocab is the vocabulary size of all annotations and # Word, # Noun, # Verb, and # Adj are the average number of words, nouns, verbs, and adjectives for each caption.

To show that our datasets reduce the many-to-many pairs, as shown in Tab 2, we report the number of classes (# cls) and instances (# inst) that involve many-to-many correspondence, indicating that our fine-grained captions largely solve the problem of a lack of distinctiveness in the previous annotations with precise video-text alignment. Counting details are in the supplementary materials.

Table 2: Many-to-many pair statistics.

|  | COG | | FIG | |
| --- | --- | --- | --- | --- |
|  | # cls | # inst | # cls | # inst |
| Charades-FIG | 1393 | 8805 | 194 | 422 |
| DiDeMo-FIG | 703 | 1925 | 32 | 65 |
| ActivityNet-FIG | 505 | 1691 | 3 | 6 |

Table 3: Detailed user study statistics.

|  | Statics | | Dynamics | | Total |
| --- | --- | --- | --- | --- | --- |
|  | $R_s(\%)$ | $S$ | $R_d(\%)$ | $S$ | $S$ |
| Charades-FIG | 100 | 4.76 | 80 | 4.38 | 4.57 |
| DiDeMo-FIG | 88 | 4.28 | 88 | 4.44 | 4.36 |
| ActivityNet-FIG | 100 | 4.44 | 84 | 4.44 | 4.44 |

To validate the accuracy of our fine-grained annotations, we conduct user studies where we randomly sample 50 statics enhanced and 50 dynamics enhanced captions for each fine-grained dataset and ask users to i) judge if our VERIFIED generated captions capture more fine-grained static or dynamic information than previous ones, and ii) grade them in 5 levels (1∼5) to measure their accuracy. We calculate the ratio of our captions $R_s$ or $R_d$ that users acknowledge to provide a richer array of static or dynamic details and report the average scores $S$ that measure accuracy. Tab 3 shows statistics, indicating that our annotations effectively extract richer, fine-grained content, with over 80% being recognized by users for their static or dynamic details. The average accuracy score of 4.46 indicates

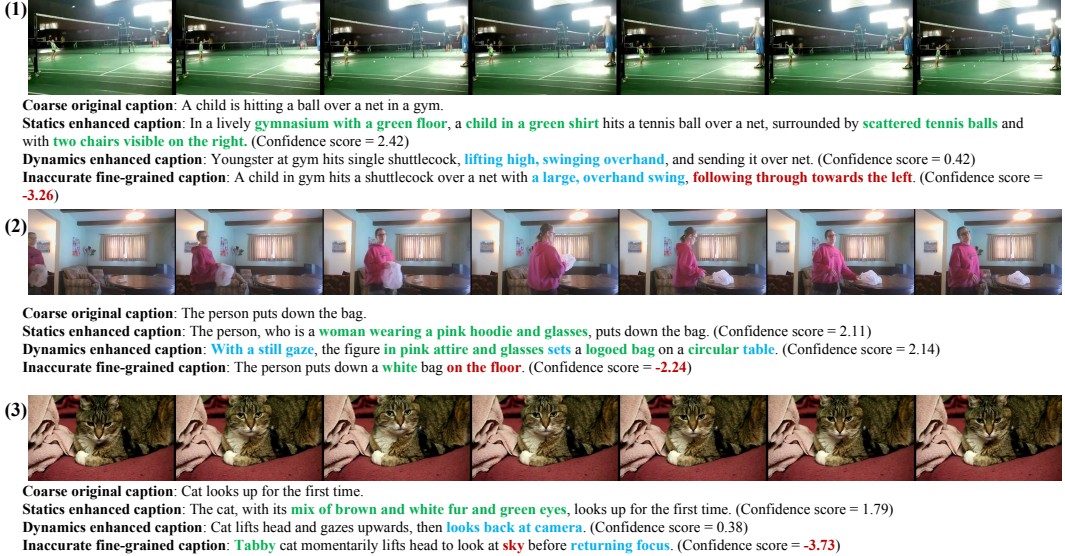

**Coarse original caption**: A child is hitting a ball over a net in a gym.
**Statics enhanced caption**: In a lively **gymnasium with a green floor**, a **child in a green shirt** hits a tennis ball over a net, surrounded by **scattered tennis balls** and with **two chairs visible on the right.** (Confidence score = 2.42)
**Dynamics enhanced caption**: Youngster at gym hits single shuttlecock, **lifting high, swinging overhand**, and sending it over net. (Confidence score = 0.42)
**Inaccurate fine-grained caption**: A child in gym hits a shuttlecock over a net with **a large, overhand swing**, **following through towards the left**. (Confidence score = **-3.26**)

**Coarse original caption**: The person puts down the bag.
**Statics enhanced caption**: The person, who is a **woman wearing a pink hoodie and glasses**, puts down the bag. (Confidence score = 2.11)
**Dynamics enhanced caption**: **With a still gaze**, the figure **in pink attire and glasses sets** a **logoed bag** on a **circular table**. (Confidence score = 2.14)
**Inaccurate fine-grained caption**: The person puts down a **white** bag **on the floor**. (Confidence score = **-2.24**)

**Coarse original caption**: Cat looks up for the first time.
**Statics enhanced caption**: The cat, with its **mix of brown and white fur and green eyes**, looks up for the first time. (Confidence score = 1.79)
**Dynamics enhanced caption**: Cat lifts head and gazes upwards, then **looks back at camera**. (Confidence score = 0.38)
**Inaccurate fine-grained caption**: **Tabby** cat momentarily lifts head to look at **sky** before **returning focus**. (Confidence score = **-3.73**)

Figure 3: Visualization of the effectiveness of our VERIFIED pipeline. (1-3) are selected from fine-grained ActivityNet-FIG, Charades-FIG, and DiDeMo-FIG, respectively. The fine-grained static and dynamic content is marked in **green** and **blue**, and inaccurate content is marked in **red**.

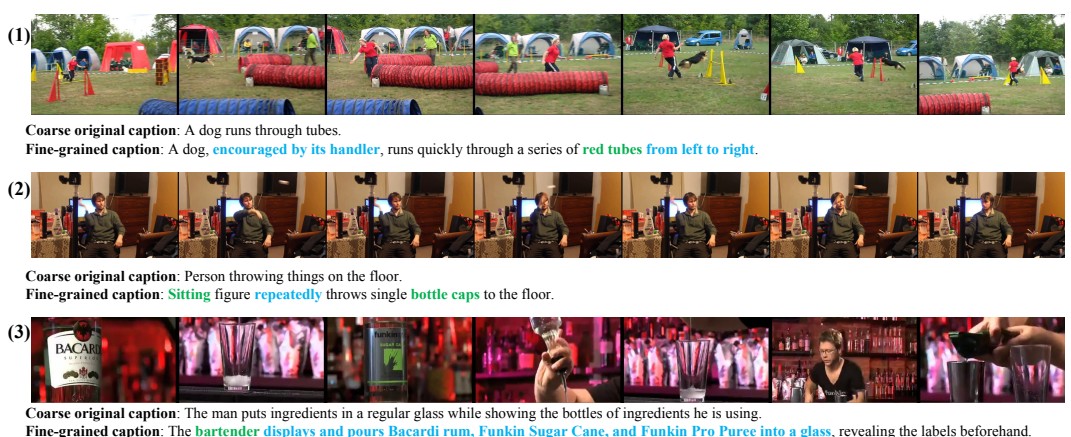

**Coarse original caption**: A dog runs through tubes.
**Fine-grained caption**: A dog, **encouraged by its handler**, runs quickly through a series of **red tubes from left to right**.

**Coarse original caption**: Person throwing things on the floor.
**Fine-grained caption**: **Sitting** figure **repeatedly** throws single **bottle caps** to the floor.

**Coarse original caption**: The man puts ingredients in a regular glass while showing the bottles of ingredients he is using.
**Fine-grained caption**: The **bartender displays and pours Bacardi rum, Funkin Sugar Cane, and Funkin Pro Puree into a glass**, revealing the labels beforehand.

Figure 4: Visualization of impressive cases. (1) Our annotation captures the interaction between the dog and its handler and movement trajectory. (2) Our annotation captures details of the throwing objects and conveys that the man throws them many times. (3) Our annotation reads the textual information from visual content and expresses the correct order of used ingredients.

high acceptance by human evaluators after filtering out inaccuracies. User study instructional texts are attached to supplementary materials.

## 3.6 Annotation Visualization

We show representative visualized samples. As shown in Fig 3, our VERIFIED pipeline reliably captures fine-grained statics and dynamics. Furthermore, annotations with inaccurate content are allocated a lower confidence score, intuitively demonstrating the effectiveness of our fine-granularity aware noise evaluator. In Fig 4, we show several successful cases, showing advantages in recognizing interaction relationships, subtle motion details, textual information, and multiple activities.

## 4 VCMR Experiment

**Methods**. We benchmark several state-of-the-art VCMR approaches: HERO [2], XML [25], Re-LoCLNet [39], CONQUER [38], SQuiDNet [41]. Among them, CONQUER and SQuiDNet need additional input, a rank list of videos of top K with scores. They retrieve moments from the top K videos other than the total video corpus. CONQUER keeps the initial rank list and scores unchanged for refined moment retrieval, while SQuiDNet learns to rerank the scores of videos for better moment retrieval. We use the video retrieval list of XML for the additional inputs of CONQUER and SQuiDNet. Implementation details and more experiments can be found in the supplementary materials.

**Metrics**. Following [3, 1], we evaluate these methods on our fine-grained VCMR datasets with VCMR, SVMR, and VR tasks. We use the $\{m\}/\text{r}\{K\}$ for VCMR and SVMR, where $m \in \{0.5, 0.7\}$ and $K \in \{1, 5, 10, 100\}$, and $\text{r}\{K\}$ for VR. $\{m\}/\text{r}\{K\}$ denotes the proportion of top $K$ proposals that have temporal Intersection over Union (tIoU) with the ground truth larger than $m$.

Table 4: Fine-grained **VCMR** and **VR** results.

| Methods | VCMR | | | | | | | | VR | | | |
|---|---|---|---|---|---|---|---|---|---|---|---|---|
| | 0.5/r1 | 0.5/r5 | 0.5/r10 | 0.5/r100 | 0.7/r1 | 0.7/r5 | 0.7/r10 | 0.7/r100 | r1 | r5 | r10 | r100 |
| | | | | | **Charades-FIG** | | | | | | | |
| HERO | 0.11 | 0.27 | 0.40 | 0.97 | 0.05 | 0.16 | 0.24 | 0.62 | 1.69 | 6.72 | 11.51 | 46.13 |
| XML | 1.05 | 2.63 | 4.33 | 9.87 | 0.43 | 1.29 | 2.26 | 5.56 | 2.80 | 8.95 | 14.11 | 51.72 |
| ReLoCLNet | 0.78 | 2.02 | 2.88 | 6.45 | 0.30 | 1.13 | 1.56 | 3.66 | 2.42 | 7.61 | 12.61 | 48.82 |
| CONQUER | 1.21 | 3.33 | 5.46 | 14.22 | 0.65 | 1.96 | 2.93 | 7.74 | 2.80 | 8.95 | 14.11 | 51.72 |
| SQuiDNet | 2.61 | 7.98 | 11.59 | 18.12 | 0.94 | 3.44 | 6.05 | 10.32 | 11.67 | 33.87 | 44.01 | 51.72 |
| | | | | | **DiDeMo-FIG** | | | | | | | |
| HERO | 0.24 | 1.34 | 1.75 | 3.83 | 0.17 | 0.77 | 1.08 | 2.28 | 8.48 | 26.73 | 39.52 | 84.46 |
| XML | 3.19 | 9.64 | 14.05 | 40.29 | 2.32 | 7.20 | 10.69 | 33.04 | 14.83 | 40.39 | 53.95 | 91.53 |
| ReLoCLNet | 3.74 | 11.01 | 15.62 | 40.29 | 1.92 | 6.75 | 9.84 | 31.47 | 14.08 | 37.18 | 50.88 | 91.30 |
| CONQUER | 5.48 | 15.45 | 22.33 | 51.63 | 3.66 | 10.12 | 15.87 | 42.64 | 14.83 | 40.39 | 53.95 | 91.53 |
| SQuiDNet | 2.89 | 7.92 | 11.94 | 33.82 | 0.52 | 1.32 | 1.99 | 6.75 | 16.94 | 44.58 | 59.26 | 91.53 |
| | | | | | **ActivityNet-FIG** | | | | | | | |
| HERO | 1.46 | 3.30 | 4.89 | 13.30 | 0.75 | 1.73 | 2.60 | 8.20 | 7.95 | 24.42 | 36.49 | 81.89 |
| XML | 2.81 | 7.86 | 12.19 | 26.28 | 1.63 | 4.58 | 7.04 | 15.24 | 13.46 | 36.37 | 49.99 | 89.31 |
| ReLoCLNet | 3.72 | 10.66 | 15.94 | 27.63 | 2.23 | 6.13 | 9.24 | 16.27 | 17.49 | 42.66 | 56.49 | 90.33 |
| CONQUER | 2.95 | 9.09 | 13.31 | 31.12 | 1.63 | 4.84 | 7.04 | 17.01 | 13.46 | 36.37 | 49.99 | 89.31 |
| SQuiDNet | 4.66 | 12.87 | 17.12 | 22.09 | 2.10 | 6.71 | 9.85 | 14.05 | 32.57 | 79.92 | 87.93 | 89.31 |

Table 5: Fine-grained **SVMR** results.

| Methods | 0.5/r1 | 0.5/r5 | 0.5/r10 | 0.5/r100 | 0.7/r1 | 0.7/r5 | 0.7/r10 | 0.7/r100 |
|---|---|---|---|---|---|---|---|---|
| | | | | **Charades-FIG** | | | | |
| HERO | 15.94 | 36.16 | 46.80 | 83.28 | 4.19 | 17.80 | 27.07 | 74.25 |
| XML | 28.20 | 61.45 | 80.27 | 95.03 | 12.90 | 34.35 | 48.31 | 67.90 |
| ReLoCLNet | 23.09 | 52.39 | 67.88 | 88.17 | 10.81 | 29.41 | 38.33 | 57.37 |
| CONQUER | 31.99 | 63.79 | 76.05 | 90.54 | 15.08 | 38.12 | 50.19 | 71.08 |
| SQuiDNet | 17.82 | 43.01 | 57.20 | 72.58 | 6.29 | 24.09 | 35.30 | 50.32 |
| | | | | **DiDeMo-FIG** | | | | |
| HERO | 9.17 | 25.01 | 32.72 | 81.08 | 3.57 | 13.87 | 21.80 | 67.83 |
| XML | 20.53 | 60.25 | 91.38 | 97.93 | 15.23 | 49.61 | 80.06 | 87.59 |
| ReLoCLNet | 22.73 | 66.11 | 88.41 | 95.74 | 11.66 | 46.87 | 78.67 | 87.49 |
| CONQUER | 34.06 | 74.53 | 95.54 | 99.28 | 20.81 | 54.05 | 82.48 | 97.38 |
| SQuiDNet | 15.28 | 41.02 | 65.24 | 76.80 | 2.12 | 7.72 | 15.03 | 41.07 |
| | | | | **ActivityNet-FIG** | | | | |
| HERO | 22.99 | 32.34 | 37.32 | 57.44 | 10.39 | 17.81 | 22.64 | 43.55 |
| XML | 25.23 | 63.19 | 72.43 | 74.14 | 13.60 | 37.81 | 44.70 | 46.02 |
| ReLoCLNet | 25.37 | 61.55 | 70.65 | 73.35 | 13.94 | 36.90 | 43.40 | 45.71 |
| CONQUER | 26.57 | 58.49 | 71.11 | 82.06 | 13.41 | 33.18 | 41.97 | 51.87 |
| SQuiDNet | 13.17 | 27.05 | 29.48 | 32.17 | 5.51 | 16.90 | 19.62 | 21.80 |

**Empirical Results**. The VCMR, VR, and SVMR results are shown in Tab 4, Tab 5. In our evaluation, XML and ReLoCLNet demonstrate comparable performance, while HERO performs the worst.

HERO utilizes a straightforward Temporal Transformer to capture correlations in video features; however, this architecture lacks the capacity to capture fine-grained relationships within videos effectively. In contrast, XML and ReLoCLNet benefit from more complicated cross-modal fusion modules, and ReLoCLNet incorporates 4 learning tasks at different granularities, which likely contributes to their superior performance.

Among the models tested, CONQUER consistently achieves strong performance across all tasks, highlighting the effectiveness of two-stage methods for VCMR. While SQuiDNet achieves the highest accuracy in VR, likely due to continued video-level learning in its second stage, it exhibits unstable performance in VCMR and lower accuracy in SVMR. Based on these observations, we recommend avoiding the entanglement of video-level and moment-level learning during the training phase in fine-grained settings. Incorporating finer-grained information during video-level retrieval learning may interfere with precise moment localization, compromising performance.

## 5 VERIFIED Pipeline Evaluation

We first explore how important our fine-grained training data is for understanding video details, to show the significance of our whole pipeline. In Tab 6, we train XML with previous coarse-grained or our fine-grained data and evaluate its performance in the fine-grained scenario. Results indicate that the impact of training with fine-grained annotations on fine-grained SVMR is relatively minor; however, it significantly enhances the performance of fine-grained VR and VCMR. This improvement is due to the fact that while fine-grained details are often redundant within a single video, they are essential for accurately pinpointing unique moments across a vast collection of similar clips. Models trained with previous coarse annotations struggle to generalize to the fine-grained scenario, especially in a large video corpus, indicating the necessity of our insight to introduce fine-grained datasets. Besides, the VCMR and VR performances on Charades-FIG are quite suboptimal, since all videos in Charades are about in-door activities and share similar semantics, making it the most challenging benchmark to evaluate methods' capability of perceiving fine-grained video differences.

Table 6: XML results with different granularities of training data.

| Training data | VCMR | | | | VR | | | | SVMR | | | |
|---|---|---|---|---|---|---|---|---|---|---|---|---|
| | 0.5/r5 | 0.5/r100 | 0.7/r5 | 0.7/r100 | r1 | r5 | r10 | r100 | 0.5/r1 | 0.5/r5 | 0.7/r1 | 0.7/r5 |
| **Charades-FIG** | | | | | | | | | | | | |
| COG | 0.89 | 3.87 | 0.43 | 2.37 | 0.62 | 2.85 | 5.75 | 30.11 | 24.73 | 57.82 | 11.80 | 32.04 |
| FIG | 2.63 | 9.87 | 1.29 | 5.56 | 2.80 | 8.95 | 14.11 | 51.72 | 28.20 | 61.45 | 12.90 | 34.35 |
| **DiDeMo-FIG** | | | | | | | | | | | | |
| COG | 6.08 | 27.81 | 4.73 | 22.35 | 8.70 | 26.84 | 38.50 | 80.89 | 17.92 | 57.14 | 11.94 | 46.05 |
| FIG | 9.64 | 40.29 | 7.20 | 33.04 | 14.83 | 40.39 | 53.95 | 91.53 | 20.53 | 60.25 | 15.23 | 49.61 |
| **ActivityNet-FIG** | | | | | | | | | | | | |
| COG | 4.41 | 17.41 | 2.48 | 10.23 | 6.70 | 21.73 | 33.63 | 80.37 | 24.31 | 60.11 | 13.04 | 37.15 |
| FIG | 7.86 | 26.28 | 4.58 | 15.24 | 13.46 | 36.37 | 49.99 | 89.31 | 25.23 | 63.19 | 13.60 | 37.81 |

Table 7: Ablation on our evaluator module using XML in VCMR task.

| | 0.5/r1 | 0.5/r5 | 0.5/r10 | 0.5/r100 | 0.7/r1 | 0.7/r5 | 0.7/r10 | 0.7/r100 |
|---|---|---|---|---|---|---|---|---|
| **Charades-FIG** | | | | | | | | |
| Lowest score | 0.67 | 1.75 | 2.85 | 8.47 | 0.32 | 0.97 | 1.64 | 4.49 |
| Highest score | 1.05 | 2.63 | 4.33 | 9.87 | 0.43 | 1.29 | 2.26 | 5.56 |
| **DiDeMo-FIG** | | | | | | | | |
| Lowest score | 2.04 | 6.30 | 10.86 | 32.67 | 1.27 | 4.46 | 8.15 | 26.31 |
| Highest score | 3.19 | 9.64 | 14.05 | 40.29 | 2.32 | 7.20 | 10.69 | 33.04 |

We show the visualization of XML in Charades-FIG when training on different granularities of training data in Fig 5. In Fig 5(b), when training on COG data, the ground truth video is out of the top 100 in its moment rank list. The top-ranked predictions mainly cover the laptop and omit other

details. In Fig 5(c), it achieves much better performance with our fine-grained data. It retrieves the target moment in rank 5, and the other candidates behind are also highly partially related to the query. It showcases the challenge of our fine-grained VCMR setting and the effectiveness of our VERIFIED generated annotations for training.

We further analyze the modules of our VERIFIED pipeline using XML [25] on the Charades-FIG and DiDeMo-FIG datasets in the VCMR task. To demonstrate the effectiveness of our evaluator, we select the caption with the highest or lowest score for each video moment to train the VCMR model. As shown in Tab 7, performance drops significantly when selecting captions with the lowest confidence scores, indicating that our evaluator can recognize better training data.

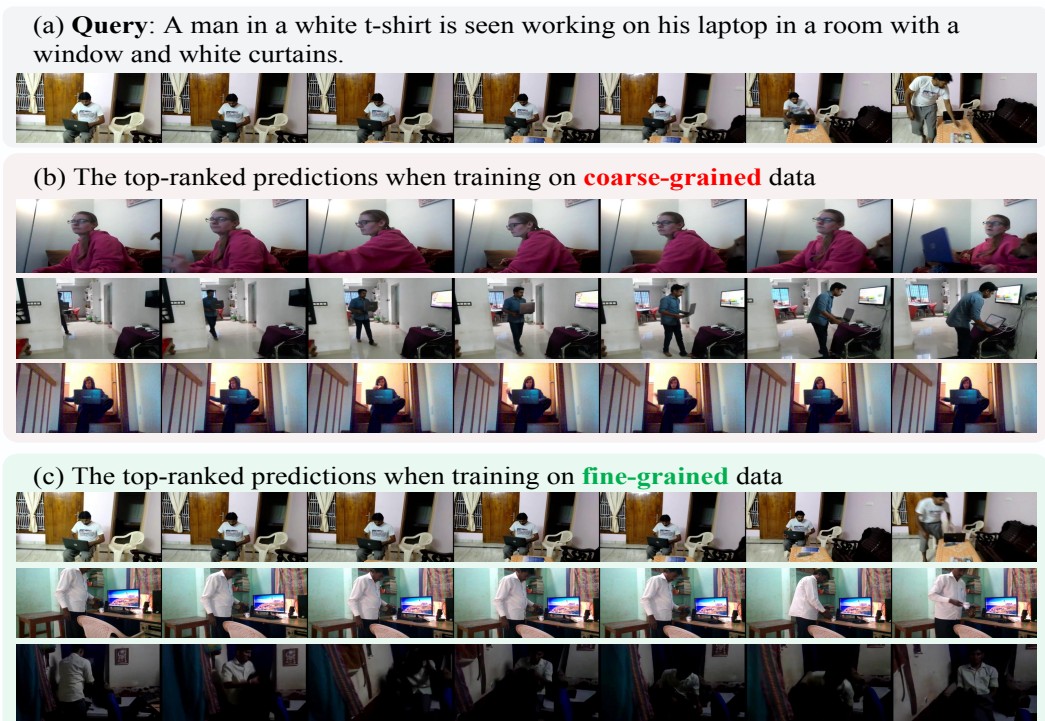

Figure 5: XML's predictions in Charades-FIG with different granularities of training data.

## 6   Conclusion and Future Work

This paper discovers that existing VCMR benchmarks' focus on coarse-grained understanding limits methods' ability to learn distinct video features and perceive fine-grained differences between video moments. Thus, we propose a more challenging fine-grained VCMR benchmark, requiring models to retrieve the best-matched moment from a video corpus given a fine-grained query, with other partially matched candidates present. To ensure efficient and high-quality video annotations, we introduce VERIFIED, an automatic video-text annotation pipeline that uses LLM and LMM to generate detailed statics and dynamics enhanced captions and filters out inaccuracies through our Fine-Granularity Aware Noise Evaluator. This evaluator is obtained via fine-tuning UMT with disturbed hard-negatives augmented contrastive and matching losses. We create the Charades-FIG, DiDeMo-FIG, and ActivityNet-FIG datasets with high-quality annotations to support fine-grained VCMR. Benchmarking state-of-the-art VCMR methods reveals that those trained on coarse annotations struggle to generalize to fine-grained scenarios, highlighting the necessity for improved fine-grained video understanding in VCMR. In the future, the next goal might be to train a completely end-to-end captioning model to complete the fine-grained annotations with the capabilities of the complicated pipeline that combines many powerful existing models. The disturbed hard negative data enhances the ability of our evaluator to understand details. However, the gap between the captioning modules' real hallucinations and our perturbation approximation does exist and it would require more analysis to reduce this gap.

## Acknowledgments and Disclosure of Funding

This work is supported by the National Key Research and Development Program of China No.2023YFF1205001, National Natural Science Foundation of China (No. 62222209, 62250008, 62102222), Beijing National Research Center for Information Science and Technology under Grant No. BNR2023RC01003, BNR2023TD03006, and Beijing Key Lab of Networked Multimedia.

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
