# VERIFIED: A Video Corpus Moment Retrieval Benchmark for Fine-Grained Video Understanding (Supplementary Materials)

**Houlun Chen**[1], **Xin Wang**[1,2*], **Hong Chen**[1], **Zeyang Zhang**[1]
**Wei Feng**[1], **Bin Huang**[1], **Jia Jia**[1,2*], **Wenwu Zhu**[1,2*]
[1] Department of Computer Science and Technology, Tsinghua University, Beijing, China
[2] BNRIST, Tsinghua University, Beijing, China
{chenhl23,h-chen20,zy-zhang20,fw22,huangb23}@mails.tsinghua.edu.cn
{xin_wang,jjia,wwzhu}@tsinghua.edu.cn
https://verified-neurips.github.io

## A  More Dataset Statistics

### A.1  Annotation Statistics

We report the confidence score distribution of our fine-grained dataset, Charades-FIG, DiDeMo-FIG, and ActivityNet-FIG, as shown in Fig 4. Most annotations have relatively high scores and ActivityNet-FIG's annotations have the highest scores. We speculate that the reason may be that the video clips in ActivityNet-FIG are relatively long, making some subtle details less important.

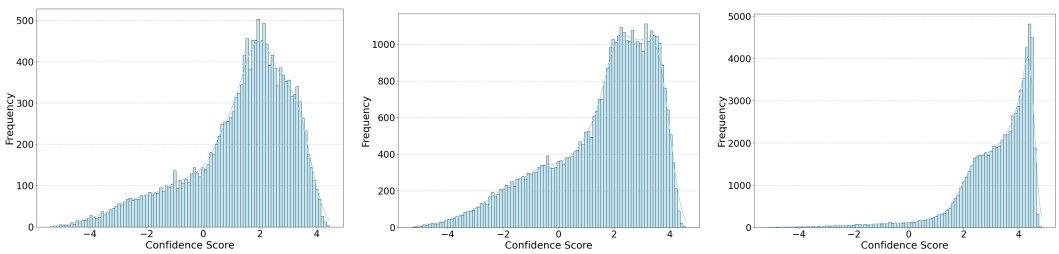

Figure 1: Charades-FIG          Figure 2: DiDeMo-FIG          Figure 3: ActivityNet-FIG

Figure 4: Confidence score distribution of the fine-grained captions.

Afterward, we report the number of annotations (Num) that are from statics enhanced candidates and dynamics enhanced candidates, respectively, and the average confidence score ($c$), as shown in Tab 1. We also list the average time span of the video moments (Span) in each dataset. For shorter video clips (Charades-FIG, DiDeMo-FIG), it's more feasible to obtain fine-grained annotations with dynamic information for their higher scores in dynamics-enhanced captions than statics-enhanced ones.

To show that our captions could be used as distinctive descriptions aligned with the video moments, that is, providing fine-grained differences between similar video clips, we conduct DBSCAN [1] clustering on the SentenceBERT embeddings of previous coarse-grained captions and our fine-grained ones. We set the DBSCAN parameters $\epsilon$ to 0.01 and the minimum instance value to 2, where we find two captions are almost identical when their cosine distance is less than 0.01. As shown in Tab 2, we report the number of classes (# class) and instances (# instance) that are grouped successfully, which

---

*Corresponding authors.

38th Conference on Neural Information Processing Systems (NeurIPS 2024) Track on Datasets and Benchmarks.

Table 1: The type and confidence score of the fine-grained captions.

| | Span/s | Statics-Enhanced | | Dynamics-Enhanced | | Total |
| | | Num | $c$ | Num | $c$ | $c$ |
|---|---|---|---|---|---|---|
| Charades-FIG | 8.23 | 11885 | 1.20 | 4243 | 1.65 | 1.32 |
| DiDeMo-FIG | 6.49 | 22703 | 1.41 | 18388 | 1.80 | 1.58 |
| ActivityNet-FIG | 37.09 | 47999 | 3.30 | 23649 | 2.90 | 3.17 |

indicates that our fine-grained captions largely solve the problem of a lack of distinctiveness in the previous annotations with precise video-text alignment.

Table 2: The clustering results of the coarse-grained and fine-grained captions.

| | COG | | FIG | |
| | # class | # instance | # class | # instance |
|---|---|---|---|---|
| Charades-FIG | 1393 | 8805 | 194 | 422 |
| DiDeMo-FIG | 703 | 1925 | 32 | 65 |
| ActivityNet-FIG | 505 | 1691 | 3 | 6 |

For intuitiveness, we demonstrate the t-SNE [2] visualization of Charades-FIG. We reset the minimum instance value to 16 for clarity. We plot the instances that are clustered in the previous coarse setting and the same ones when they are annotated with fine-grained captions, shown in Fig 5.

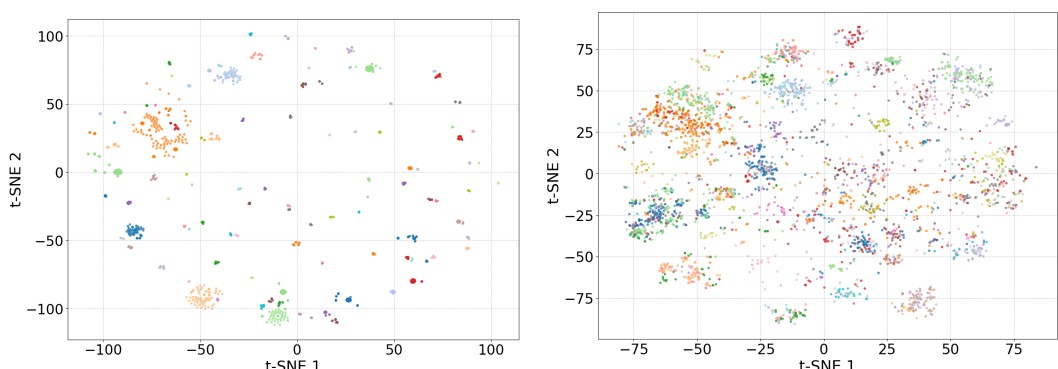

(a) Clustering on the previous coarse-grained annotations of Charades-STA.

(b) Clustering on our fine-grained annotations of Charades-FIG.

Figure 5: Clustering visualization for annotations on Charades-STA and Charades-FIG.

## A.2 Detailed User Study Statistics

As stated in the full paper, to validate the accuracy of our fine-grained annotations, we conduct user studies where we randomly sample 50 statics enhanced and 50 dynamics enhanced captions for each fine-grained dataset and ask users to i) judge if our VERIFIED generated captions capture more fine-grained static or dynamic information than previous ones with "Yes" or "No", and ii) grade them in 5 levels ($1 \sim 5$) to measure their accuracy. Each question is answered by three different participants. Please refer to detailed instructions in Sec D. We calculate the ratio of our captions $R_s(\%)$ or $R_d(\%)$ that users acknowledge to provide a richer array of static or dynamic details and report the average scores $S$ that measure accuracy. Higher metrics indicate better alignment with humans. The statistics are in Tab 3. Both statics-enhanced and dynamics-enhanced captions achieve comparable levels of user acceptance with similar accuracy.

Table 3: Detailed user study statistics.

| | Statics-Enhanced | | Dynamics-Enhanced | | Total |
| --- | --- | --- | --- | --- | --- |
| | $R_s(\%)$ | $S$ | $R_d(\%)$ | $S$ | $S$ |
| Charades-FIG | 100 | 4.76 | 80 | 4.38 | 4.57 |
| DiDeMo-FIG | 88 | 4.28 | 88 | 4.44 | 4.36 |
| ActivityNet-FIG | 100 | 4.44 | 84 | 4.44 | 4.44 |

Table 4: Fine-grained **VCMR** results with different training granularity.

| | train | test | 0.5/r1 | 0.5/r5 | 0.5/r10 | 0.5/r100 | 0.7/r1 | 0.7/r5 | 0.7/r10 | 0.7/r100 |
| --- | --- | --- | --- | --- | --- | --- | --- | --- | --- | --- |
| **Charades-FIG** | | | | | | | | | | |
| HERO | COG | FIG | 0.03 | 0.05 | 0.13 | 0.30 | 0.00 | 0.05 | 0.05 | 0.19 |
| | FIG | FIG | 0.11 | 0.27 | 0.40 | 0.97 | 0.05 | 0.16 | 0.24 | 0.62 |
| | COG | COG | 0.00 | 0.00 | 0.16 | 0.32 | 0.00 | 0.05 | 0.11 | 0.22 |
| | FIG | COG | 0.00 | 0.05 | 0.22 | 0.30 | 0.00 | 0.00 | 0.03 | 0.05 |
| XML | COG | FIG | 0.46 | 0.89 | 1.45 | 3.87 | 0.19 | 0.43 | 0.83 | 2.37 |
| | FIG | FIG | 1.05 | 2.63 | 4.33 | 9.87 | 0.43 | 1.29 | 2.26 | 5.56 |
| | COG | COG | 0.43 | 0.94 | 1.53 | 5.27 | 0.30 | 0.62 | 0.99 | 3.06 |
| | FIG | COG | 0.48 | 1.05 | 1.83 | 4.06 | 0.24 | 0.46 | 1.10 | 2.58 |
| ReLoCLNet | COG | FIG | 0.24 | 0.56 | 0.97 | 2.82 | 0.05 | 0.22 | 0.43 | 1.26 |
| | FIG | FIG | 0.78 | 2.02 | 2.88 | 6.45 | 0.30 | 1.13 | 1.56 | 3.66 |
| | COG | COG | 0.27 | 0.54 | 0.81 | 2.18 | 0.13 | 0.27 | 0.35 | 1.05 |
| | FIG | COG | 0.32 | 0.83 | 1.29 | 3.17 | 0.19 | 0.46 | 0.83 | 1.91 |
| **DiDeMo-FIG** | | | | | | | | | | |
| HERO | COG | FIG | 0.07 | 0.22 | 0.37 | 1.54 | 0.02 | 0.07 | 0.17 | 0.75 |
| | FIG | FIG | 0.24 | 1.34 | 1.75 | 3.83 | 0.17 | 0.77 | 1.08 | 2.28 |
| | COG | COG | 0.02 | 0.41 | 0.65 | 1.32 | 0.02 | 0.12 | 0.41 | 0.81 |
| | FIG | COG | 0.12 | 0.32 | 0.42 | 1.10 | 0.05 | 0.15 | 0.27 | 0.65 |
| XML | COG | FIG | 1.94 | 6.08 | 9.62 | 27.81 | 1.35 | 4.73 | 7.43 | 22.35 |
| | FIG | FIG | 3.19 | 9.64 | 14.05 | 40.29 | 2.32 | 7.20 | 10.69 | 33.04 |
| | COG | COG | 1.52 | 4.49 | 7.00 | 22.00 | 1.02 | 3.41 | 5.46 | 18.37 |
| | FIG | COG | 1.17 | 3.71 | 6.03 | 21.18 | 0.87 | 2.74 | 4.86 | 17.34 |
| ReLoCLNet | COG | FIG | 1.69 | 5.31 | 8.30 | 26.24 | 0.75 | 2.77 | 4.93 | 19.96 |
| | FIG | FIG | 3.74 | 11.01 | 15.62 | 40.29 | 1.92 | 6.75 | 9.84 | 31.47 |
| | COG | COG | 1.54 | 4.46 | 7.10 | 20.81 | 0.85 | 2.79 | 4.71 | 15.50 |
| | FIG | COG | 1.12 | 3.91 | 6.83 | 19.91 | 0.55 | 2.27 | 4.16 | 14.33 |
| **ActivityNet-FIG** | | | | | | | | | | |
| HERO | COG | FIG | 0.68 | 2.04 | 3.28 | 9.86 | 0.31 | 1.05 | 1.70 | 5.73 |
| | FIG | FIG | 1.46 | 3.30 | 4.89 | 13.30 | 0.75 | 1.73 | 2.60 | 8.20 |
| | COG | COG | 0.60 | 1.68 | 2.58 | 8.62 | 0.34 | 1.03 | 1.57 | 5.36 |
| | FIG | COG | 0.54 | 1.39 | 2.09 | 6.12 | 0.30 | 0.84 | 1.28 | 3.98 |
| XML | COG | FIG | 1.37 | 4.41 | 7.19 | 17.41 | 0.84 | 2.48 | 4.16 | 10.23 |
| | FIG | FIG | 2.81 | 7.86 | 12.19 | 26.28 | 1.63 | 4.58 | 7.04 | 15.24 |
| | COG | COG | 0.95 | 3.70 | 6.38 | 14.68 | 0.56 | 2.19 | 3.78 | 8.95 |
| | FIG | COG | 0.95 | 3.16 | 5.17 | 13.67 | 0.60 | 1.92 | 3.12 | 8.25 |
| ReLoCLNet | COG | FIG | 1.51 | 5.25 | 8.33 | 17.01 | 0.93 | 2.91 | 4.73 | 10.01 |
| | FIG | FIG | 3.72 | 10.66 | 15.94 | 27.63 | 2.23 | 6.13 | 9.24 | 16.27 |
| | COG | COG | 1.11 | 3.88 | 6.32 | 12.45 | 0.68 | 2.31 | 3.93 | 8.07 |
| | FIG | COG | 1.08 | 3.60 | 6.01 | 13.25 | 0.67 | 2.26 | 3.65 | 8.15 |

Table 5: Fine-grained **SVMR** results with different training granularity.

| | train | test | 0.5/r1 | 0.5/r5 | 0.5/r10 | 0.5/r100 | 0.7/r1 | 0.7/r5 | 0.7/r10 | 0.7/r100 |
|---|---|---|---|---|---|---|---|---|---|---|
| **Charades-FIG** | | | | | | | | | | |
| HERO | COG | FIG | 18.25 | 38.23 | 47.88 | 83.04 | 5.11 | 18.20 | 27.98 | 74.27 |
| | FIG | FIG | 15.94 | 36.16 | 46.80 | 83.28 | 4.19 | 17.80 | 27.07 | 74.25 |
| | COG | COG | 18.06 | 38.92 | 48.66 | 83.31 | 5.00 | 18.28 | 27.90 | 74.6 |
| | FIG | COG | 17.53 | 36.80 | 46.94 | 83.09 | 5.24 | 18.41 | 27.12 | 74.25 |
| XML | COG | FIG | 24.73 | 57.82 | 77.69 | 94.17 | 11.80 | 32.04 | 46.37 | 65.54 |
| | FIG | FIG | 28.20 | 61.45 | 80.27 | 95.03 | 12.90 | 34.35 | 48.31 | 67.90 |
| | COG | COG | 31.40 | 64.22 | 81.21 | 95.81 | 13.82 | 35.38 | 48.17 | 66.94 |
| | FIG | COG | 30.86 | 63.31 | 80.62 | 95.51 | 15.00 | 36.21 | 48.87 | 67.12 |
| ReLoCLNet | COG | FIG | 22.39 | 48.87 | 66.42 | 89.33 | 10.91 | 28.44 | 37.96 | 57.58 |
| | FIG | FIG | 23.09 | 52.39 | 67.88 | 88.17 | 10.81 | 29.41 | 38.33 | 57.37 |
| | COG | COG | 26.21 | 55.27 | 69.27 | 88.20 | 11.05 | 29.73 | 38.68 | 56.85 |
| | FIG | COG | 23.04 | 51.72 | 67.04 | 87.82 | 10.81 | 28.90 | 37.74 | 56.10 |
| **DiDeMo-FIG** | | | | | | | | | | |
| HERO | COG | FIG | 9.17 | 23.47 | 31.47 | 79.97 | 3.44 | 11.89 | 18.42 | 64.96 |
| | FIG | FIG | 9.17 | 25.01 | 32.72 | 81.08 | 3.57 | 13.87 | 21.80 | 67.83 |
| | COG | COG | 10.08 | 25.63 | 33.92 | 81.22 | 4.22 | 14.61 | 22.66 | 67.93 |
| | FIG | COG | 9.69 | 23.70 | 31.60 | 78.67 | 4.01 | 13.26 | 19.89 | 65.09 |
| XML | COG | FIG | 17.92 | 57.14 | 90.26 | 98.13 | 11.94 | 46.05 | 79.24 | 87.84 |
| | FIG | FIG | 20.53 | 60.25 | 91.38 | 97.93 | 15.23 | 49.61 | 80.06 | 87.59 |
| | COG | COG | 21.58 | 61.60 | 91.40 | 98.18 | 15.95 | 51.18 | 80.86 | 87.54 |
| | FIG | COG | 21.03 | 60.38 | 91.45 | 97.98 | 15.60 | 49.94 | 80.49 | 87.24 |
| ReLoCLNet | COG | FIG | 22.45 | 64.84 | 87.09 | 97.06 | 9.89 | 45.73 | 77.55 | 87.29 |
| | FIG | FIG | 22.73 | 66.11 | 88.41 | 95.74 | 11.66 | 46.87 | 78.67 | 87.49 |
| | COG | COG | 23.92 | 67.38 | 88.69 | 97.38 | 11.84 | 46.97 | 77.90 | 87.62 |
| | FIG | COG | 22.23 | 66.18 | 88.24 | 95.74 | 11.11 | 46.10 | 78.79 | 87.37 |
| **ActivityNet-FIG** | | | | | | | | | | |
| HERO | COG | FIG | 22.84 | 33.37 | 39.11 | 58.65 | 10.24 | 18.01 | 23.53 | 45.18 |
| | FIG | FIG | 22.99 | 32.34 | 37.32 | 57.44 | 10.39 | 17.81 | 22.64 | 43.55 |
| | COG | COG | 22.56 | 35.24 | 41.57 | 60.26 | 10.24 | 19.03 | 25.32 | 46.82 |
| | FIG | COG | 21.02 | 31.45 | 36.99 | 56.50 | 9.24 | 16.80 | 21.78 | 42.62 |
| XML | COG | FIG | 24.31 | 60.11 | 66.76 | 67.35 | 13.04 | 37.15 | 42.64 | 43.22 |
| | FIG | FIG | 25.23 | 63.19 | 72.43 | 74.14 | 13.60 | 37.81 | 44.70 | 46.02 |
| | COG | COG | 28.74 | 68.34 | 74.76 | 76.05 | 15.72 | 43.65 | 48.31 | 49.37 |
| | FIG | COG | 27.17 | 65.77 | 73.89 | 75.30 | 14.50 | 40.65 | 46.94 | 48.10 |
| ReLoCLNet | COG | FIG | 23.18 | 57.50 | 65.65 | 67.54 | 12.75 | 34.26 | 40.43 | 41.97 |
| | FIG | FIG | 25.37 | 61.55 | 70.65 | 73.35 | 13.94 | 36.90 | 43.40 | 45.71 |
| | COG | COG | 29.09 | 68.35 | 73.59 | 75.11 | 16.04 | 43.54 | 47.34 | 48.74 |
| | FIG | COG | 27.34 | 65.47 | 72.29 | 74.68 | 14.74 | 40.17 | 45.26 | 47.17 |

# B    More VCMR Experiments

## B.1    Implementation Details

In the section, we report more detailed results of XML [3] and ReLoCLNet [4] and HERO [5] with different granularities training and test data. For important hyper-parameters, we set the clip length to 1, 1, 1, maximum prediction length to 12, 30, 180, minimum prediction length to 2, 5, 1, maximum context length to 48, 30, 240, and nms threshold to 0.6, 0.6, 0.5, for Charades-FIG, DiDeMo-FIG, ActivityNet-FIG. For ActivityNet-FIG, we evaluate models on val_2 splits. For a fair comparison, we extract RoBERTa [6] text features and extract 2048D ResNet-152 [7] video features, where we first

Table 6: Fine-grained **VR** results with different training granularity.

| | train | test | r1 | r5 | r10 | r100 |
|---|---|---|---|---|---|---|
| | | | **Charades-FIG** | | | |
| HERO | COG | FIG | 0.40 | 2.45 | 4.44 | 24.89 |
| | FIG | FIG | 1.69 | 6.72 | 11.51 | 46.13 |
| | COG | COG | 0.67 | 2.07 | 3.92 | 27.15 |
| | FIG | COG | 0.48 | 2.39 | 4.35 | 21.99 |
| XML | COG | FIG | 0.62 | 2.85 | 5.75 | 30.11 |
| | FIG | FIG | 2.80 | 8.95 | 14.11 | 51.72 |
| | COG | COG | 0.83 | 2.82 | 5.48 | 27.31 |
| | FIG | COG | 0.86 | 2.90 | 5.22 | 27.96 |
| ReLoCLNet | COG | FIG | 0.75 | 2.69 | 4.92 | 28.74 |
| | FIG | FIG | 2.42 | 7.61 | 12.61 | 48.82 |
| | COG | COG | 0.54 | 2.58 | 4.76 | 26.67 |
| | FIG | COG | 0.78 | 2.77 | 4.84 | 26.08 |
| | | | **DiDeMo-FIG** | | | |
| HERO | COG | FIG | 3.99 | 13.56 | 22.28 | 72.09 |
| | FIG | FIG | 8.48 | 26.73 | 39.52 | 84.46 |
| | COG | COG | 2.97 | 11.11 | 19.04 | 63.76 |
| | FIG | COG | 3.02 | 11.51 | 18.86 | 58.29 |
| XML | COG | FIG | 8.70 | 26.84 | 38.50 | 80.89 |
| | FIG | FIG | 14.83 | 40.39 | 53.95 | 91.53 |
| | COG | COG | 5.61 | 17.87 | 26.94 | 68.53 |
| | FIG | COG | 5.38 | 16.90 | 26.07 | 66.41 |
| ReLoCLNet | COG | FIG | 7.40 | 22.45 | 32.57 | 80.54 |
| | FIG | FIG | 14.08 | 37.18 | 50.88 | 91.30 |
| | COG | COG | 5.68 | 17.69 | 25.99 | 66.26 |
| | FIG | COG | 5.36 | 16.00 | 24.89 | 65.49 |
| | | | **ActivityNet-FIG** | | | |
| HERO | COG | FIG | 3.35 | 12.60 | 20.88 | 66.85 |
| | FIG | FIG | 7.95 | 24.42 | 36.49 | 81.89 |
| | COG | COG | 2.62 | 10.15 | 16.89 | 57.52 |
| | FIG | COG | 2.47 | 9.47 | 15.77 | 53.04 |
| XML | COG | FIG | 6.70 | 21.73 | 33.63 | 80.37 |
| | FIG | FIG | 13.46 | 36.37 | 49.99 | 89.31 |
| | COG | COG | 4.06 | 14.65 | 24.20 | 66.42 |
| | FIG | COG | 3.74 | 13.40 | 21.40 | 62.31 |
| ReLoCLNet | COG | FIG | 7.20 | 22.77 | 34.34 | 78.31 |
| | FIG | FIG | 17.49 | 42.66 | 56.49 | 90.33 |
| | COG | COG | 4.10 | 14.95 | 23.51 | 65.28 |
| | FIG | COG | 4.44 | 14.88 | 23.76 | 62.47 |

extract features of 4fps sampled video frames and apply max-pooling every 1 second. Specifically, for ReLoCLNet, since we do not use subtitles, we replace the cross-attention feature fusion between subtitles and videos with self-attention on video features. Other training details are adhere to the official codebases.

## B.2 Full Experimental Results

We report more detailed experimental results in Tab 4, 5, 6, where COG/FIG represents training or test on coarse-grained/fine-grained annotations. There are more observations.

1. Focusing on the "FIG FIG" lines, XML performs the best in Charades-FIG and ReLoCLNet performs the best in ActivityNet-FIG across VCMR and VR tasks, suggesting that XML is better in perceiving subtle video content and ReLoCLNet is more robust to longer videos.

2. Comparing the "COG COG" and "FIG COG" lines across these tables, we find that training on the fine-grained annotations only slightly impacts the performance in the coarse-grained scenario in all tasks compared to training on the coarse-grained ones. Considering that coarse-granularity training struggles to generalize to the fine-grained VCMR and VR scenarios, we suggest that it's a better option to replace coarse-granularity training with fine-granularity training since its high quality and stronger generalization to different scenarios.

## C   More VERIFIED Pipeline Evaluation

## D   Instructional Texts for User Study

Here are the instructional texts for the user study. To illustrate clearly, we have also posted an example of our user study in the dir "user_study_example". We pay each participant 5 dollars on average.

Listing 1: The instructional texts for the user study

```
1
2  ### User Study for Video Fine-Grained Annotation
3
4  #### General Description
5
6  This experiment consists of 30 evaluation questions, aiming to assess
       ↪ the fine-grained descriptive ability of Description 2: users
       ↪ need to evaluate whether Description 2 provides more **rich**
       ↪ and **accurate** **dynamic** and **static** fine-grained
       ↪ descriptive information compared to Description 1.
7
8  There are two types of descriptive information:
9
10 1. **Static Information:** This mainly refers to information in the
       ↪ video that does not change significantly over time, which can
       ↪ be obtained from static video frames, such as environment,
       ↪ scenes, and attributes of objects (e.g., names, colors, sizes,
       ↪ text information appearing in the video, gender and age of
       ↪ people, positional relationships between objects, etc.).
11 2. **Dynamic Information:** This mainly refers to information in the
       ↪ video that changes over time and can only be understood by
       ↪ watching the complete video (e.g., changes in positions,
       ↪ lighting, camera angles, objects appearing or disappearing at
       ↪ certain moments, new behaviors or activities, interaction
       ↪ relationships, action details, etc.).
12
13 As shown in Sample 1 (please open the HTML file), in the statics, "A
       ↪ dog runs quickly through a series of tubes in the grass with
       ↪ some camps nearby," the phrases "in the grass" and "some camps
       ↪ nearby" do not change over time and can be obtained from a few
       ↪ static video frames, thus considered static information. In the
       ↪  dynamics, "A dog encouraged by its handler runs quickly
       ↪ through a series of red tubes from left to right," the phrases
       ↪ "encouraged by its handler," "runs through red tubes," and "
       ↪ from left to right" involve interaction relationships and
       ↪ positional changes, which vary over time, thus considered
       ↪ dynamic information.
14
15 The concepts of statics and dynamics are not completely opposite but
       ↪ have different focuses. Users should answer based on their
       ↪ understanding. The samples are for reference only. Users only
       ↪ need to focus on the content appearing in Description 2 and do
       ↪ not need to worry about whether Description 2 fully and
       ↪ comprehensively describes the video.
```

```
16
17  #### Instructions for Questions
18
19  For each question, a video clip is provided along with Description 1
        ↪  and Description 2. Users need to evaluate:
20
21  1. Whether Description 2, compared to Description 1, introduces richer
        ↪   **static** fine-grained information.
22
23     The answer is represented by 0 or 1. 1 indicates **introduces**
            ↪  richer static fine-grained information, and 0 indicates **
            ↪  does not introduce** richer static fine-grained information.
24
25  2. Whether Description 2, compared to Description 1, introduces richer
        ↪   **dynamic** fine-grained information.
26
27     The answer is represented by 0 or 1. 1 indicates **introduces**
            ↪  richer dynamic fine-grained information, and 0 indicates **
            ↪  does not introduce** richer dynamic fine-grained information
            ↪  .
28
29  3. Evaluate the accuracy of Description 2 in fine-grained description.
30
31     The answer should be a number from [1, 2, 3, 4, 5], where:
32
33     - 5: The description is perfectly accurate, or there are some vague
            ↪   areas in the description but due to the video's content, it
            ↪   is still acceptable to humans.
34     - 4: The description has some inappropriate information, but the
            ↪  importance of the inappropriate content is low, thus the
            ↪  impact is minor.
35     - 3: The description has some inappropriate information, causing a
            ↪  certain degree of impact.
36     - 2: The description has many errors, but still maintains
            ↪  consistency with the video at a coarse level.
37     - 1: The description has serious errors and is completely unrelated
            ↪   to the video.
38
39  #### Note:
40
41  After completing all questions, **fill in your student ID** in the
        ↪  corresponding location to facilitate the issuance of subsidies.
```

## E   LLM/LMM Prompts

We take the prompts used for constructing ActivityNet-FIG for example.

Listing 2: The prompt for generating static $\mathcal{D}_i^{(fg)}$ $\mathcal{D}_i^{(bg)}$ and $\mathcal{D}_i$

```
1
2  This image is extracted from a video clip that describes "{query}".
        ↪  Now you are asked to supplement the description with detailed
        ↪  attributes of this image, including background, and other
        ↪  detailed attributes of the main focused objects if they appear
        ↪  in this image. You may focus on age, color, quantity,
        ↪  positional relationships, and so on if such attributes are
        ↪  clear. You should only say what you observed and not give
        ↪  suggestions or indications. Pay attention that you should
        ↪  answer it briefly. Answer with the format:
3
4  "Background: ...,
5  Objects Attributes:..."
6
```

```
7  Finally, generate a new brief description mainly based on the original
   ↪  description with attributes information incorporated. The
   ↪  narratives should be similar to the original description.
   ↪  Answer with the format:
8
9  "Desc:...".
```

Listing 3: The prompt for generating statics enhanced caption candidates $\{q_i^{(s)}\}_{i=1}^{N_s}$

```
1
2  Augment the caption of the video clip with fine-grained static content
   ↪  (scenes, objects, background...). You are given detailed
   ↪  descriptions of several keyframes in the video clip. What you
   ↪  need to do is find significant and reliable information that is
   ↪   missing in the original caption and enrich the original
   ↪  caption.
3
4  This is an example:
5
6  Suppose you find some new significant information, you can answer 3
   ↪  different candidates in JSON format:
7
8  {{
9      "original": " Another man is interviewed by the camera and shows
          ↪  off his surfing skills on his board.",
10
11     "1": "Another man is interviewed by the camera in front of a
          ↪  surfboard, wearing a red cap and yellow shirt, and shows
          ↪  off his surfing skills on his board, skillfully navigating
          ↪  large, breaking waves in the ocean with another surfer in
          ↪  the distance.",
12
13     "2": "Another man is interviewed by the camera, standing before a
          ↪  surfboard with a Red Bull cap and yellow shirt, and shows
          ↪  off his surfing skills on his board, riding dynamic ocean
          ↪  waves during the VANS WORLD CUP OF SURFING.",
14
15     "3": "Another man is interviewed by the camera, sporting a red cap
          ↪   and yellow shirt, and shows off his surfing skills on his
          ↪  board, expertly maneuvering through frothy waves in the
          ↪  ocean, with a backdrop of a surfing event sign and a
          ↪  bicycle nearby."
16 }}
17
18 Now it's your turn to finish the task.
19
20 The original caption of this video clip is "{original_description}"
   ↪  There are {keyframe_number} keyframes with detailed information
   ↪   in temporal order:
21
22 {attribute_content}
23
24 You enrich the original caption based on the given keyframe
   ↪  descriptions and answer in the format below which contains at
   ↪  least 3 enriched captions.
25
26 "
27 {{
28     "original": "...",
29
30     "1": "...",
31
32     "2": "...",
33
```

```
34        "3": "..."
35 }}
36 "
37 The enhanced caption should be brief in one sentence and similar to
     ↪ the original one in style and length. Details in the original
     ↪ caption should not be missed!
```

Listing 4: The prompt for generating dynamics-oriented questions $\{\mathcal{Q}_i\}_{i=1}^{N_{qa}}$

```
1
2 There is a coarse description of a video clip that shows an activity.
     ↪ An image understanding expert has told you all the static and
     ↪ spatial information about this video to you, like the object
     ↪ and the scene. Now you want to know dynamic and temporal
     ↪ information about this clip. Suppose you can ask a video
     ↪ understanding expert questions about this video and it will
     ↪ tell you.
3
4 For example, the original description is "groups of people are dancing
     ↪ .", and you can ask "What are the detailed moves of this dance
     ↪ ?"
5 For example, again, the original description is "a man rides a bicycle
     ↪  in front of a house.", and you can ask "What is the direction
     ↪ of the man's moving?"
6
7 Ok, the current video description is "{original_description}". You
     ↪ should prepare at least 5 questions about the dynamic and
     ↪ temporal content of the video based on the original description
     ↪ , focusing on the action or motion itself. You can use common
     ↪ knowledge when you think about it. Generate your answer briefly
     ↪  with one sentence for each question in the format of "1. ...
     ↪ 2. ...  3. ...   4. ... 5. ...".
```

Listing 5: The prompt for generating dynamics-oriented answers and descriptions $\{\mathcal{A}_i\}_{i=1}^{N_{qa}} \tilde{\mathcal{D}}$

```
1
2 The video has a duration of {duration} seconds and is sampled to {
     ↪ frame_number} frames. The frames are ordered and indexed by
     ↪ 1,2,3... This video clip shows "{description}"(an original
     ↪ description). You should briefly answer the questions one-by-
     ↪ one in one sentence in the format of "1. ... 2. ... 3. ...".
3
4 {questions}
5
6 Then you give a detailed description of the dynamic information in
     ↪ this video, e.g. actions and motions. You can take into
     ↪ consideration the variation of position, size, light, camera
     ↪ shot, and other activities that are not described in the
     ↪ original description, and so on. You can say more details of
     ↪ the people's actions. If there are other dynamic objects, you
     ↪ can additionally describe them in a sentence.
```

Listing 6: The prompt for generating dynamics enhanced caption candidates $\{q_i^{(d)}\}_{i=1}^{N_d}$

```
1
2 An original caption describes a specific moment in a video. Now you
     ↪ should modify the original caption by adding more fine-grained
     ↪ temporally dynamic information. There are several questions
     ↪ about the dynamic details and a strong expert answers these
     ↪ questions and gives a detailed description of the dynamic info
     ↪ in this video moment. You should refer to this information and,
     ↪  importantly, preserve the original semantics. You should not
     ↪ make up fake content. You are NOT required to utilize all
```

```
           ↪ dynamic information and you should consider some distinctive
           ↪ and reliable dynamic information, like the variation of the
           ↪ position, and more activities and interactions, finally form a
           ↪ new short and brief caption in one sentence like the original
           ↪ one in writing styles. You rewrite the new caption, following
           ↪ the example's format.
3
4  Here is an example, suppose the original caption is "All three dancer
       ↪ have their backs to the camera." You find more dynamic
       ↪ information and you can output like:
5
6  "All three dancer have their backs to the camera and move in
       ↪ synchronized circles."
7
8  Now it's your turn to finish this task.
9
10 An original caption "{original}" describes a specific moment, from {st
       ↪ } to {ed} seconds, in a video.
11
12 Questions and Answers:
13 1.
14 (Q) {q1}
15 (A) {a1}
16
17 2.
18 (Q) {q2}
19 (A) {a2}
20
21 3.
22 (Q) {q3}
23 (A) {a3}
24
25 4.
26 (Q) {q4}
27 (A) {a4}
28
29 5.
30 (Q) {q5}
31 (A) {a5}
32
33
34 Dynamic description:
35 {dynamic_description}
36
37 Your rewrited caption should be only in one sentence and brief,
       ↪ similar to the original caption in writing style, with as FEW
       ↪ words as possible. You should NOT make up any unexisting
       ↪ content and you should reserve the original words! You should
       ↪ hide words about explicit time information because this will
       ↪ leak clues! They both should be short in words. You should
       ↪ consider some distinctive and reliable dynamic information,
       ↪ like variation of the position, and more activities and
       ↪ interactions. They should all be authentic and fine-grained but
       ↪  brief. You give about 3 DIFFERENT candidate answers in the
       ↪ format "1. ... 2. ... 3. ..."
```

Listing 7: The prompt for generating positively rewritten captions $\{q_i^+\}_{i=1}^{N_{pos}}$

```
1
2  You are required to rewrite the sentence with the same meanings and
       ↪ similar styles and lengths. Use commonly used simple words and
       ↪ phrases!
3
4  For example, the original sentence is:
```

```
 5   "A little boy stands up to chase a cat on the grass."
 6   You can answer:
 7   1. "A boy stands up to run after a cat on the grass."
 8   2. "A little boy gets up to chase a cat on the grass."
 9   3. "A little boy stands up, and then chases a kitty on the grass."
10
11   For example, the original sentence is:
12   "A yellow automobile arrives and stops beside a house."
13   You can answer:
14   1. "There is a yellow car arriving and stopping beside a house."
15   2. "A yellow car pulls up and parks beside a house."
16   3. "A yellow automobile arrives and stops next to a house."
17
18   Now following these examples, it's your turn. The original sentence is
         ↪ :
19   "{original}".
20   Now you give your rewritten sentence, 3 different candidates, NOT
         ↪ change the meanings, in the format of 1. ... 2. ... 3. ...:
```

Listing 8: The prompt for generating statics disturbed negative captions $\{q_i^{s-}\}_{i=1}^{N_{neg}}$

```
 1
 2
 3   You are required to rewrite the sentence to change some STATIC fine-
         ↪ grained information (like color, age, position, new objects or
         ↪ persons, light, background, and so on) while maintaining other
         ↪ meanings, to make the rewriting one distinctive to the original
         ↪  (a negtive sample of the original one). Use commonly used
         ↪ simple words and phrases with similar writing styles and
         ↪ lengths!
 4
 5   For example, the original sentence is:
 6   "A little boy stands up to run after a cat on the grass."
 7   You can answer:
 8   1. "A little girl stands up to run after a cat on the grass."
 9   2. "A boy stands up to run after a dog on the grass."
10   3. "A little boy stands up to run after a cat on the street."
11
12   For example, the original sentence is:
13   "A yellow car pulls up and parks beside the left window of the house."
14   You can answer:
15   1. "A white car pulls up and parks beside the left window of the house
         ↪ ."
16   2. "A yellow car pulls up and parks beside the right window of the
         ↪ house."
17   3. "A yellow bus pulls up and parks between two houses."
18
19   Now following these examples, it's your turn. The original sentence is
         ↪ :
20   "{original}".
21   Now you give your rewritten brief sentence, 3 different candidates, to
         ↪  modify important fine-grained static details to vary semantics
         ↪ , remaining others unchanged, in the format of 1. ... 2. ... 3.
         ↪  ...:
```

Listing 9: The prompt for generating dynamics disturbed negative captions $\{q_i^{d-}\}_{i=1}^{N_{neg}}$

```
 1
 2   You are required to rewrite the sentence to change some DYNAMIC fine-
         ↪ grained information (like size variation, direction or position
         ↪  variation, light variation, modifying or adding motion and
         ↪ activities, new interactions, and so on) while maintaining
         ↪ other meanings, to make the rewriting one distinctive to the
         ↪ original (a negtive sample of the original one). Use commonly
```

```
     ↪ used simple words and phrases with similar writing styles and
     ↪ lengths!
3
4  For example, the original sentence is:
5  "A little boy stands up to run after a cat on the grass."
6  You can answer:
7  1. "A little boy stops running after a cat on the grass."
8  2. "A cat runs after a little boy on the grass."
9  3. "A little boy turns back to look at a cat on the grass."
10
11 For example, the original sentence is:
12 "A yellow car passes the cross from left to right."
13 You can answer:
14 1. "A yellow car finally stops in front of the cross."
15 2. "A yellow car passes the cross from right to left."
16 3. "A yellow car goes past people along the street."
17
18 Now following these examples, it's your turn. The original sentence is
     ↪ :
19 "{original}".
20 Now you give your rewritten brief sentence, 3 different candidates, to
     ↪  modify important fine-grained dynamic details, remaining
     ↪ others unchanged, in the format of 1. ... 2. ... 3. ...:
```