# OpenReview forum: "VERIFIED: A Video Corpus Moment Retrieval Benchmark for Fine-Grained Video Understanding"
_NeurIPS.cc/2024/Datasets_and_Benchmarks_Track — NeurIPS 2024 Track Datasets and Benchmarks Poster_

### Official Review · Reviewer_r7iW · 2024-07-24
**A novel annotation pipeline and comprehensive VCMR benchmarks via LLMs and LMMs**

**Rating:** 7
**Confidence:** 3
**Correctness:** Yes.
**Clarity:** Yes.

**Review:**

In this paper, the authors highlight the importance of fine-grained video moment retrieval which relies on the detailed textual description. The proposed VCMR pipeline leverages LLMs and LMMs to build both static and dynamic enhanced captions, which is clearly explained in the paper and the figures. Also, the noise evaluator is really impressive based on simple but effective perturbations to train a model as a filter, which highly improves the data quality and mitigate the hallucination. However, there's still a concern on the pipeline. The capability of the evaluators are highly related with the perturbed data, which may have different distribution with the real LLM hallucination. Also, if I think this evaluation only works for the static part and is not available for the dynamic captions.

Also, this pipeline is applied to three popular video datasets, with clear experimental details in the paper and supplementary material. Also, a few VCMR models are evaluated on these datasets, showing the limitation of these models on fine-grained datasets. However, the analysis is much more about the general observations. It should be better if the paper can cover detailed analysis about the reason of the low performance of each model, either from the perspective of model structure or from the view of data.

Pros:
1. An annotation pipeline to re-build fine-grained VCMR datasets via LLMs and LMMs, with a noise evaluator to remove hallucinations
2. Three coarse-grained datasets are converted to fine-grained datasets, with benchmarking results on baseline models.

Cons:
1. The reliability of the evaluation model is not guaranteed. It lacks a direct hallucination analysis or a human spot checking;
2. The evaluation to the dynamic enhanced captions is missing;
3. There are only general observations to the benchmarking results, which makes this paper less helpful on how to improve the performance of the model on fine-grained datasets in the future.

**Strengths:**

See Pros in Review

**Additional Feedback:**

I hope the authors can address my concern in the rebuttal, especially those about the pipeline.

**Documentation:**

Yes.

**Ethics:**

No.

**Limitations:**

Not very adequate. Only one sentence in the conclusion.

**Opportunities For Improvement:**

See Cons in Review

**Relation To Prior Work:**

Yes.

**Summary And Contributions:**

This paper proposes a novel benchmark and annotation pipeline to improve fine-grained VCMR. First, the authors propose VERIFIED, an annotation pipeline based on LLMs and LMMs to generate detailed captions with both static an dynamic enhanced captions. Then the reconstructed datasets are evaluated with some baseline VCMR models.

---

> ### Author Rebuttal · Authors · 2024-08-18
>
> **1. A concern on the pipeline. The capability of the evaluator is highly related to the perturbed data, which may have a different distribution from the real LLM hallucination.**
>
> Thanks for your insightful suggestion. Our manual assessment verifies the high quality of annotation and the perturbed data could be seen to have an approximate distribution to our annotation system's hallucination, empirically, where the perturbed data covers common errors and helps the system to recognize them. Besides, our fine-granularity-aware fine-tuning aims to enhance its ability to focus on detailed information in the query. Our evaluator is fine-tuned based on UMT, which has been pretrained on large-scale data, equipping it with the generalization ability on various objects and activities, which also helps to mitigate the influence of the distribution gap.
>
> We conduct a quantitative ablation experiment to better show the capability of the evaluator. We introduce the ablation setting, where we select the caption with the lowest score. We conduct this part with XML in the VCMR task.
>
> | Charades-FIG | 0.5/r1 | 0.5/r5 | 0.5/r10 | 0.5/r100 | 0.7/r1 | 0.7/r5 | 0.7/r10 | 0.7/r100 |
> | ------------ | ------ | ------ | ------- | -------- | ------ | ------ | ------- | -------- |
> | Low score    | 0.67   | 1.75   | 2.85    | 8.47     | 0.32   | 0.97   | 1.64    | 4.49     |
> | High score   | 1.05   | 2.63   | 4.33    | 9.87     | 0.43   | 1.29   | 2.26    | 5.56     |
>
>
> | DiDeMo-FIG | 0.5/r1 | 0.5/r5 | 0.5/r10 | 0.5/r100 | 0.7/r1 | 0.7/r5 | 0.7/r10 | 0.7/r100 |
> | ---------- | ------ | ------ | ------- | -------- | ------ | ------ | ------- | -------- |
> | Low score  | 2.04   | 6.30   | 10.86   | 32.67    | 1.27   | 4.46   | 8.15    | 26.31    |
> | High score | 3.19   | 9.64   | 14.05   | 40.29    | 2.32   | 7.20   | 10.69   | 33.04    |
>
> We can see when selecting captions with the lowest confidence scores, the performance drops a lot, which indicates our evaluator is capable of recognizing better training data.
>
> Besides, we conduct a deeper human assessment to evaluate the capability of the evaluator, where we randomly sample 100 annotation pairs, each of which contains the highest-score and lowest-score one for a video moment, and ask users to evaluate their qualities. Given users a video moment and two captions, users are asked to select one of the four cases:
>
> * Sentence 1 has some inaccurate content.
> * Sentence 2 has some inaccurate content.
> * Both sentences have some inaccurate content.
> * Both sentences are correct.
>
> | The high score caption has inaccuracy | The low score caption has inaccuracy | Both have inaccuracy | Both are correct |
> | ------------------------------------- | ------------------------------------ | -------------------- | ---------------- |
> | 3                                     | 19                                   | 3                    | 75               |
>
> 94% of the captions selected by our evaluator are correct enough to be accepted by humans. Only 78% of the captions are correct when we choose the caption with the lowest score. Such ablation proves the effectiveness of our evaluator.
>
> The distribution gap is inherently present. It might be challenging to evaluate the real hallucination distribution of LLM/LVMs. Thus, it remains a new topic to provide a theoretical or statistical approach to analyze this gap to estimate the bound of hallucination. For example, annotating the inaccurate phrases on the LLM/LVM's output, which could serve as a statistical distribution of errors, might help to fine-tune LLM to generate better-perturbed data. We will add this discussion to the paper later.
>
> **2. I think this evaluation only works for the static part and is not available for the dynamic captions.**
>
> Thanks for your concern. The evaluation works for dynamic captions as well. The reasons are:
>
> (i) The foundation UMT (Paper: Unmasked Teacher: Towards Training-Efficient Video Foundation Models) model is pretrained with action recognition, action detection tasks, etc to handle scene-related and temporal-related actions and conduct complex video-language understanding. It gives an example of instructing the model to distinguish the actions of "opening" and "closing", as shown in Figure 2 of this paper(https://openaccess.thecvf.com/content/ICCV2023/papers/Li_Unmasked_Teacher_Towards_Training-Efficient_Video_Foundation_Models_ICCV_2023_paper.pdf).
>
> (ii) In our fine-granularity aware fine-tuning stage, we disturb the queries to vary the dynamic content. It can utilize the abilities acquired by the model during the pretraining stage and guide the model to focus on crucial dynamic content.
>
> There are some examples of our disturbed queries.
>
> * Charades-FIG
>
>   Desc: A person is opening a box.
>
>   Dynamic Disturbed: A person shakes the box before opening it.
>
> * DiDeMo-FIG
>
>   Desc: The second man from the right brings his arrow close to his bow.
>
>   Dynamic Disturbed: The man second from the right pulls back his arrow and releases it from the bow.
>
> * ActivityNet-FIG
>
>   Desc: Unfold the bottom part of the tissue paper to create a flower appearance.
>
>   Dynamic Disturbed: Unfold the top of the tissue paper to reveal a hidden design.
>
> Some cases demonstrate our evaluator can capture the hallucination of dynamics and give a relatively low matching score.
>
> Figure 3(1) in the main paper serves as an example. There are other examples in this rebuttal PDF. We give the scores of captions. The dynamic inaccurate content is marked in red. We can see that it can recognize hallucinations like left/right, shot movement, etc.

---

> ### Author Rebuttal · Authors · 2024-08-18
>
> **3. It would be better if the paper could cover a detailed analysis of the reason for the low performance of each model, either from the perspective of model structure or from the view of data. Observations are limited, making this paper less helpful on how to improve the performance of the model in fine-grained settings.**
>
> Thanks for your suggestion. Thanks for this suggestion. We add deeper analysis of the benchmark datasets to analyze reasons for some models' performances and reveal some laws and directions to develop future video corpus moment retrieval models. To enhance the persuasiveness of the analysis, we add CONQUER and SQuiDNet(\* in the table), and they are second-stage-focused methods relying on some first-stage outputs. Compared to others, they need additional input, a rank list of videos of top K with scores. They retrieve moments from the top K videos other than the total video corpus. CONQUER keeps the initial rank list and scores *unchanged* for refined moment retrieval, while SQuiDNet introduces a loss function to learn to *rerank* the scores of videos for better moment retrieval. We use the video retrieval list of XML for the additional inputs of CONQUER and SQuiDNet.
>
> We list the FIG-FIG setting in Charades-FIG and DiDeMo-FIG in the tasks of VCMR and VR.
>
> * Charades-FIG, VCMR
>
> | Method     | 0.5/r1 | 0.5/r5 | 0.5/r10 | 0.5/r100 | 0.7/r1 | 0.7/r10 | 0.7/r50 | 0.7/r100 |
> | ---------- | ------ | ------ | ------- | -------- | ------ | ------- | ------- | -------- |
> | HERO       | 0.11   | 0.27   | 0.40    | 0.97     | 0.05   | 0.16    | 0.24    | 0.62     |
> | XML        | 1.05   | 2.63   | 4.33    | 9.87     | 0.43   | 1.29    | 2.26    | 5.56     |
> | ReLoCLNet  | 0.78   | 2.02   | 2.88    | 6.45     | 0.30   | 1.13    | 1.56    | 3.66     |
> | CONQUER\*  | 1.21   | 3.33   | 5.46    | 14.22    | 0.65   | 1.96    | 2.93    | 7.74     |
> | SQuiDNet\* | 2.61   | 7.98   | 11.59   | 18.12    | 0.94   | 3.44    | 6.05    | 10.32    |
>
> * Charades-FIG, VR
>
> | Method     | r1    | r5    | r10   | r100  |
> | ---------- | ----- | ----- | ----- | ----- |
> | HERO       | 1.69  | 6.72  | 11.51 | 46.13 |
> | XML        | 2.80  | 8.95  | 14.11 | 51.72 |
> | ReLoCLNet  | 2.42  | 7.61  | 12.61 | 48.82 |
> | CONQUER\*  | 2.80  | 8.95  | 14.11 | 51.72 |
> | SQuiDNet\* | 11.67 | 33.87 | 44.01 | 51.72 |
>
> * DiDeMo-FIG, VCMR
>
> | Method     | 0.5/r1 | 0.5/r5 | 0.5/r10 | 0.5/r100 | 0.7/r1 | 0.7/r10 | 0.7/r50 | 0.7/r100 |
> | ---------- | ------ | ------ | ------- | -------- | ------ | ------- | ------- | -------- |
> | HERO       | 0.24   | 1.34   | 1.75    | 3.83     | 0.17   | 0.77    | 1.08    | 2.28     |
> | XML        | 3.19   | 9.64   | 14.05   | 40.29    | 2.32   | 7.20    | 10.69   | 33.04    |
> | ReLoCLNet  | 3.74   | 11.01  | 15.62   | 40.29    | 1.92   | 6.75    | 9.84    | 31.47    |
> | CONQUER\*  | 5.48   | 15.45  | 22.33   | 51.63    | 3.66   | 10.12   | 15.87   | 42.64    |
> | SQuiDNet\* | 2.89   | 7.92   | 11.94   | 33.82    | 0.52   | 1.32    | 1.99    | 6.75     |
>
> * DiDeMo-FIG, VR
>
> | Method     | r1    | r5    | r10   | r100  |
> | ---------- | ----- | ----- | ----- | ----- |
> | HERO       | 8.48  | 26.73 | 39.52 | 84.46 |
> | XML        | 14.83 | 40.39 | 53.95 | 91.53 |
> | ReLoCLNet  | 14.08 | 37.18 | 50.88 | 91.30 |
> | CONQUER\*  | 14.83 | 40.39 | 53.95 | 91.53 |
> | SQuiDNet\* | 16.94 | 44.58 | 59.26 | 91.53 |
>
> XML and ReLoCLNet show similar performances and HERO is the worst. HERO only applies a simple Temporal Transformer to model the correlations in the video features and such architecture is not capable of learning the fine-grained relations within videos. XML and ReLoCLNet have many more parameters in cross-modal fusion modules and ReLoCLNet designs 4 learning tasks in various granularities, which support their better performances.
>
> Overall, CONQUER is the best-performing method. We recommend two-stage methods for VCMR. Although SQuiDNet achieves the best in VR because it continues video-level learning in the second stage, it shows *unstable* performance in VCMR. **We suggest not entangling video-level and moment-level learning** in the training stage for the fine-grained setting. The reason is that introducing more fine-grained information when learning video-level retrieval might interfere with moment localization.
>
> We will add this analysis to the paper later. Thank you.

---

> ### Author Rebuttal · Authors · 2024-08-18
>
> **4. The reliability of the evaluation model is not guaranteed. It lacks a direct hallucination analysis or human spot-checking.**
>
> Thanks for your suggestion. We have conducted a manual check of our generated data and the quality is accepted by human evaluators(see details in Section B.2 of supplementary materials `supplementary_materials.pdf`). We will move it to the main paper. We conduct additional ablation to study the reliability of the evaluation model.
>
> We first conduct a quantitative ablation experiment. We introduce the ablation setting, where we select the caption with the lowest score. We conduct this part with XML in the VCMR task.
>
> | Charades-FIG | 0.5/r1 | 0.5/r5 | 0.5/r10 | 0.5/r100 | 0.7/r1 | 0.7/r5 | 0.7/r10 | 0.7/r100 |
> | ------------ | ------ | ------ | ------- | -------- | ------ | ------ | ------- | -------- |
> | Low score    | 0.67   | 1.75   | 2.85    | 8.47     | 0.32   | 0.97   | 1.64    | 4.49     |
> | High score   | 1.05   | 2.63   | 4.33    | 9.87     | 0.43   | 1.29   | 2.26    | 5.56     |
>
>
> | DiDeMo-FIG | 0.5/r1 | 0.5/r5 | 0.5/r10 | 0.5/r100 | 0.7/r1 | 0.7/r5 | 0.7/r10 | 0.7/r100 |
> | ---------- | ------ | ------ | ------- | -------- | ------ | ------ | ------- | -------- |
> | Low score  | 2.04   | 6.30   | 10.86   | 32.67    | 1.27   | 4.46   | 8.15    | 26.31    |
> | High score | 3.19   | 9.64   | 14.05   | 40.29    | 2.32   | 7.20   | 10.69   | 33.04    |
>
> We can see when selecting captions with the lowest confidence scores, the performance drops a lot, which indicates our evaluator is capable of recognizing better training data.
>
> Besides, we then conduct a deeper human assessment, where we randomly sample 100 annotation pairs, each of which contains the highest-score and lowest-score one for a video moment, and ask users to evaluate their qualities. Given users a video moment and two captions, users are asked to select one of the four cases:
>
> * Sentence 1 has some inaccurate content.
> * Sentence 2 has some inaccurate content.
> * Both sentences have some inaccurate content.
> * Both sentences are correct.
>
> | The high score caption has inaccuracy | The low score caption has inaccuracy | Both have inaccuracy | Both are correct |
> | ------------------------------------- | ------------------------------------ | -------------------- | ---------------- |
> | 3                                     | 19                                   | 3                    | 75               |
>
> 94% of the captions selected by our evaluator are correct enough to be accepted by humans. Only 78% of the captions are correct when we choose the caption with the lowest score. Such ablation proves the reliability of our evaluator.
>
> We will add this part to the paper.
>
> **5. The evaluation of the dynamic enhanced captions is missing.**
>
> We have user evaluation for both statics and dynamics enhanced captions in the supplementary materials(`supplementary_materials.pdf`) in Section B.2 (Line 54 - 65). In Table 3, there are 80%, 88%, and 84% of the dynamics-enhanced captions are accepted by users to have more detailed dynamic information in Charades-FIG, DiDeMo-FIG, and ActivityNet-FIG. The average scores are 4.38, 4.44, and 4.44, showing high quality for dynamics-enhanced captions. We will move it to the main paper.
>
> **6. The limitation is not very adequate.**
>
> Thanks for your suggestion. The discussion about modeling the distribution gap between the real hallucination of our pipeline and the augmented disturbed data is interesting and in-depth. We will add it to the paper.

---

> > ### Comment · Reviewer_r7iW · 2024-08-23
> > **Response to Authors**
> >
> > I appreciate your new experiments and they somehow resolved my concerns. I would like to raise my score to 7.

---

### Official Review · Reviewer_G9CT · 2024-07-24
**The paper introduces a novel benchmark for Video Corpus Moment Retrieval (VCMR) that focuses on fine-grained video understanding.**

**Rating:** 4
**Confidence:** 4
**Correctness:** Yes
**Clarity:** No.

**Review:**

Pros
1. The paper introduces an automated pipeline in detail, which uses LLMs and LMMs to create detailed video annotations. In addition, to alleviate the hallucination of LLMs, it proposes a noise evaluator to enhance accuracy.
2. Based on the above pipeline, the paper introduces a novel and more challenging VCMR benchmark concentrate more on fine-grained video features, which is expected to be valuable to the community.

Cons
1. The paper does not provide evaluation results by enough ablation studies. The author only give the final result and ignore to reveal how much role each component in the pipeline plays in the final evaluation result. e.g., Statics Enhanced Captioning, Dynamics Enhanced Captioning, Fine-Granularity Aware Noise Evaluator……
2. The article lacks some qualitative analysis or visualization of the test samples, which allows readers to more intuitively feel how this solution focuses on the fine-grained features of the video.
3. There are some problems with formatting. Tables 3 and 4 on page 9 should follow Table 2 instead of being inserted in the middle of Chapter 5: conclusion.

**Strengths:**

1. The paper introduces an automated pipeline in detail, which uses LLMs and LMMs to create detailed video annotations. In addition, to alleviate the hallucination of LLMs, it proposes a noise evaluator to enhance accuracy.
2. Based on the above pipeline, the paper introduces a novel and more challenging VCMR benchmark concentrate more on fine-grained video features, which is expected to be valuable to the community.

**Additional Feedback:**

Please refer to the weaknesses and questions part.

**Documentation:**

Yes.

**Ethics:**

No.

**Limitations:**

Same as the weakness.

**Opportunities For Improvement:**

1. Can you supply more experiment results to support the validity of your paper？Especially the ablation studies of each component.
2. It would be beneficial to generate visualization for the test samples to understand the fine-grained capability of the benchmark. This information may be included in the Appendix if necessary.

**Relation To Prior Work:**

Yes.

**Summary And Contributions:**

The paper introduces a novel benchmark for Video Corpus Moment Retrieval (VCMR) that focuses on fine-grained video understanding. The authors argue that existing VCMR benchmarks are limited to coarse-grained understanding, which hampers the precise localization of video moments when given fine-grained queries. To address this limitation, the paper proposes a more challenging VCMR scenario that requires models to retrieve the best-matched video moment from a corpus with other partially matched candidates.
The paper's main contributions are as follows:
1. It proposes a challenging scenario requiring precise video moment localization in response to fine-grained queries amidst similar candidates.
2. It introduces an automated pipeline using LLMs and LMMs to create detailed video annotations with a noise evaluator to enhance accuracy. And it develops high-quality datasets (Charades-FIG, DiDeMo-FIG, ActivityNet-FIG) to facilitate fine-grained VCMR.
4. It demonstrates the inadequacy of existing models when evaluated on the new datasets, underlining the importance of fine-grained understanding in VCMR.

---

> ### Author Rebuttal · Authors · 2024-08-18
>
> **1. Ablations are not enough.**
>
> We conduct quantitative ablation experiments to support the effectiveness of our captioning modules. We set a baseline annotation pipeline using Gemini-1.5-Pro with vanilla prompts to get captions to capture fine-grained objects, actions, etc. We train the XML model with such vanilla captions. Here are the results.
>
> | Charades-FIG | 0.5/r1 | 0.5/r5 | 0.5/r10 | 0.5/r100 | 0.7/r1 | 0.7/r5 | 0.7/r10 | 0.7/r100 |
> | ------------ | ------ | ------ | ------- | -------- | ------ | ------ | ------- | -------- |
> | Vanilla      | 0.51   | 1.40   | 2.39    | 6.02     | 0.22   | 0.65   | 1.02    | 3.23     |
> | Our          | 1.05   | 2.63   | 4.33    | 9.87     | 0.43   | 1.29   | 2.26    | 5.56     |
>
> We can see that the performance drops significantly when replacing our training data with vanilla ones, showing that getting fine-grained captions through our statics- and dynamics-enhanced captioning modules is important.
>
> We then show some qualitative cases to show the differences between the captions from vanilla prompts and our captioning modules. There are examples in this rebuttal PDF (Figure 1,2,3).
>
> We could see, that without our statics and dynamics enhanced modules, the vanilla version lacks fine-grained information like rich character attributes, environments, and motion details. Some even have errors, like "makes a pfft sound" in the Figure 3.
>
> To better demonstrate the effectiveness of our evaluator. We conduct a quantitative ablation experiment. We introduce the ablation setting, where we select the caption with the lowest score. We conduct this part with XML in the VCMR task.
>
> | Charades-FIG | 0.5/r1 | 0.5/r5 | 0.5/r10 | 0.5/r100 | 0.7/r1 | 0.7/r5 | 0.7/r10 | 0.7/r100 |
> | ------------ | ------ | ------ | ------- | -------- | ------ | ------ | ------- | -------- |
> | Low score    | 0.67   | 1.75   | 2.85    | 8.47     | 0.32   | 0.97   | 1.64    | 4.49     |
> | High score   | 1.05   | 2.63   | 4.33    | 9.87     | 0.43   | 1.29   | 2.26    | 5.56     |
>
>
> | DiDeMo-FIG | 0.5/r1 | 0.5/r5 | 0.5/r10 | 0.5/r100 | 0.7/r1 | 0.7/r5 | 0.7/r10 | 0.7/r100 |
> | ---------- | ------ | ------ | ------- | -------- | ------ | ------ | ------- | -------- |
> | Low score  | 2.04   | 6.30   | 10.86   | 32.67    | 1.27   | 4.46   | 8.15    | 26.31    |
> | High score | 3.19   | 9.64   | 14.05   | 40.29    | 2.32   | 7.20   | 10.69   | 33.04    |
>
> We can see when selecting captions with the lowest confidence scores, the performance drops a lot, which indicates our evaluator is capable of recognizing better training data.
>
> Besides, we conduct a deeper human assessment, where we randomly sample 100 annotation pairs, each of which contains the highest-score and lowest-score one for a video moment, and ask users to evaluate their qualities. Given users a video moment and two captions, users are asked to select one of the four cases:
>
> * Sentence 1 has some inaccurate content.
> * Sentence 2 has some inaccurate content.
> * Both sentences have some inaccurate content.
> * Both sentences are correct.
>
> | The high score caption has inaccuracy | The low score caption has inaccuracy | Both have inaccuracy | Both are correct |
> | ------------------------------------- | ------------------------------------ | -------------------- | ---------------- |
> | 3                                     | 19                                   | 3                    | 75               |
>
> 94% of the captions selected by our evaluator are correct enough to be accepted by humans. Only 78% of the captions are correct when we choose the caption with the lowest score. Such ablation proves the effectiveness of our evaluator.
>
> Furthermore, when both captions are correct, the caption selected by our evaluator often has richer information and is more expressive (Figure 4 of this rebuttal PDF).
>
> We will add this part to the paper. Thank you!

---

> ### Author Rebuttal · Authors · 2024-08-18
>
> **2. It lacks some qualitative analysis of the test samples.**
>
> We visualize some test samples of the current models tested on FIG when trained on COG or FIG. Refer to the figures in this rebuttal PDF.
>
> We select XML in Charades-FIG for visualization. In this case, the fine-grained query text is "A man in a white t-shirt is seen working on his laptop in a room with a window and white curtains.", the ground truth is Figure 1. We evaluate the model in the FIG test set. In Figure 2, when the model is trained on the previous COG setting, the ground truth video is out of the top 100 in its moment rank list. The top-ranked predictions mainly cover the laptop and omit other details. It showcases the challenge of our FIG test benchmark. In Figure 3, while it’s trained on the FIG training set, it achieves much better performance. It retrieves the target moment in rank 5, and the other candidates behind are also highly partially related to the query. The moment in the 2nd line has a computer, white clothes, and curtains although the curtains are not white and the computer is not strictly a laptop. The moment in the 3rd line has laptops and white clothes.
>
> Figure 4 in this rebuttal PDF showcases a failure case. The model is trained and tested in FIG. The query is "A man in a white t-shirt is seen working on his laptop in a room with a window and white curtains". When trained in the FIG training set, the model’s rank 7 prediction covers white clothes but not the laptop. Sometimes, users focus more on laptops other than white clothes.
>
> With more detailed information and a larger vocabulary, the challenge lies there’s a much larger potentially relevant candidate pool, making it harder to retrieve target moments precisely. The model sometimes focuses on some content and neglects others.
>
> We will add this part to the main paper later.
>
> **3. Formatting problems.**
>
> Thanks. We would adjust the layout of the paper.

---

> > ### Comment · Area_Chair_eh1D · 2024-08-26
> >
> > Can Reviewer G9CT please comment on the authors' rebuttal, both on qualitative analysis and on ablation studies? Thank you.

---

### Official Review · Reviewer_i5C5 · 2024-07-24
**Review of submission #246**

**Rating:** 7
**Confidence:** 3
**Correctness:** Yes
**Clarity:** Yes

**Review:**

- The experimental section is too shallow. How is the benchmark challenging for current models ? What are the failure cases ? What do you recommend for developing future models ? The goal of a new benchmark is to provide a signal for improving models and this part is not discussed at all in the paper. One suggestion for the presentation, the tables take a lot of the space, their size could be reduced to save space for an analysis and discussion of the results.

- One of the goals of having more fine-grained captions is also to reduce the uncertainty when searching a given moment described by text in a large corpus of video. Indeed, the number of clips corresponding to the given caption should drop to 1. It would be nice to have a study on how close you are to being fully deterministic, e.g, for a given caption in your dataset, how many corresponding clips do you have in average in the corpus and how is this number improved from previous benchmarks. More generally, it would be nice to have an analysis on how much harder this benchmark is compared to previous ones.

- If the captions are automatically generated by LLMs (say Gemini), then the performance of a model that would completely solve the benchmark is bounded by what these LLMs are capable of ? Or bounded by engineered approaches that combine all the models used (PySceneDetect, Gemini, Mistral ect). Is the goal then to obtain a completely end-to-end model that would have the capabilities of all these models combined ? Do you have thoughts about that ?

**Strengths:**

- The benchmark is a step change in terms of difficulty for VCMR systems. There is a real improvement in the granularity of the captions compared to existing benchmarks, and the results show that the benchmark is actually challenging for current models, which is always a good property for a new benchmark.

- The automatic captioning pipeline is interesting. In particular the Fine-Granularity Aware Noise Estimator is a novel component that seems to work very well. Something that is not discussed is that the pipeline as it is has potential for fine-grained captioning applications. Moreover, the pipeline is well-described with lots of implementation details allowing reproducibility.

**Additional Feedback:**

No

**Documentation:**

Yes

**Limitations:**

Yes

**Opportunities For Improvement:**

See review.

**Relation To Prior Work:**

Yes

**Summary And Contributions:**

This paper presents a new Video Copus Moment Retrieval (VCMR) benchmark that is significantly more fine-grained than existing ones. From existing VCMR datasets, they propose an automatic procedure to enhance the captions and make them much more descriptive, both for visual content description and for temporal events descriptions. They finally evaluate current best approaches such as XML and ReLoCLNet on the proposed benchmark.

---

> ### Author Rebuttal · Authors · 2024-08-18
>
> **1. The experimental section is shallow.**
>
> > How is the benchmark challenging for current models?
>
> We would answer this question from the perspectives of **statistics** and **visualizations**. (i) We count the number of queries with more than 1 potential positive video moment in both COG and FIG that fails to have the model learn the golden matching, referring to Table 2 in the supplementary materials(`supplementary_materials.pdf`). We will move this part to the main paper later. Our FIG benchmark provides more one-to-one matching cases which requires the model to retrieve the best matching one. Besides, when new details are introduced, the vocabulary enlarges, referring to Table 1 of the main paper, a larger vocabulary with more descriptive words also brings challenges. (ii) We visualize some cases of the current models tested on FIG when trained on COG or FIG to show the challenge. Refer to the figures in this rebuttal PDF.
>
> We select XML in Charades-FIG for visualization. In this case, the fine-grained query text is "A man in a white t-shirt is seen working on his laptop in a room with a window and white curtains.", the ground truth is Figure 1. We evaluate the model in the FIG test set. In Figure 2, when the model is trained on the previous COG setting, the ground truth video is out of the top 100 in its moment rank list. The top-ranked predictions mainly cover the laptop and omit other details. It showcases the challenge of our FIG test benchmark. In Figure 3, while it’s trained on the FIG training set, it achieves much better performance. It retrieves the target moment in rank 5, and the other candidates behind are also highly partially related to the query. The moment in the 2nd line has a computer, white clothes, and curtains although the curtains are not white and the computer is not strictly a laptop. The moment in the 3rd line has laptops and white clothes.
>
> > What are the failure cases?
>
> Figure 4 in this rebuttal PDF showcases a failure case. The model is trained and tested in FIG. The query is "A man in a white t-shirt is seen working on his laptop in a room with a window and white curtains". When trained in the FIG training set, the model’s rank 7 prediction covers white clothes but not the laptop. Sometimes, users focus more on laptops other than white clothes.
>
> With more detailed information and a larger vocabulary, the challenge lies there’s a much larger potentially relevant candidate pool, making it harder to retrieve target moments precisely. The model sometimes focuses on some content and neglects others.

---

> ### Author Rebuttal · Authors · 2024-08-18
>
> > What do you recommend for developing future models? The goal of a new benchmark is to provide a signal for improving models and this part is not discussed at all in the paper.
>
> Thanks for this suggestion. We add deeper analysis of the benchmark datasets to reveal some laws and directions to develop future video corpus moment retrieval models. To enhance the persuasiveness of the analysis, we add CONQUER and SQuiDNet(\* in the table), and they are second-stage-focused methods relying on some first-stage outputs. Compared to others, they need additional input, a rank list of videos of top K with scores. They retrieve moments from the top K videos other than the total video corpus. CONQUER keeps the initial rank list and scores *unchanged* for refined moment retrieval, while SQuiDNet introduces a loss function to learn to *rerank* the scores of videos for better moment retrieval. We use the video retrieval list of XML for the additional inputs of CONQUER and SQuiDNet.
>
> We list the FIG-FIG setting in Charades-FIG and DiDeMo-FIG in the tasks of VCMR and VR.
>
> * Charades-FIG, VCMR
>
> | Method     | 0.5/r1 | 0.5/r5 | 0.5/r10 | 0.5/r100 | 0.7/r1 | 0.7/r10 | 0.7/r50 | 0.7/r100 |
> | ---------- | ------ | ------ | ------- | -------- | ------ | ------- | ------- | -------- |
> | HERO       | 0.11   | 0.27   | 0.40    | 0.97     | 0.05   | 0.16    | 0.24    | 0.62     |
> | XML        | 1.05   | 2.63   | 4.33    | 9.87     | 0.43   | 1.29    | 2.26    | 5.56     |
> | ReLoCLNet  | 0.78   | 2.02   | 2.88    | 6.45     | 0.30   | 1.13    | 1.56    | 3.66     |
> | CONQUER\*  | 1.21   | 3.33   | 5.46    | 14.22    | 0.65   | 1.96    | 2.93    | 7.74     |
> | SQuiDNet\* | 2.61   | 7.98   | 11.59   | 18.12    | 0.94   | 3.44    | 6.05    | 10.32    |
>
> * Charades-FIG, VR
>
> | Method     | r1    | r5    | r10   | r100  |
> | ---------- | ----- | ----- | ----- | ----- |
> | HERO       | 1.69  | 6.72  | 11.51 | 46.13 |
> | XML        | 2.80  | 8.95  | 14.11 | 51.72 |
> | ReLoCLNet  | 2.42  | 7.61  | 12.61 | 48.82 |
> | CONQUER\*  | 2.80  | 8.95  | 14.11 | 51.72 |
> | SQuiDNet\* | 11.67 | 33.87 | 44.01 | 51.72 |
>
> * DiDeMo-FIG, VCMR
>
> | Method     | 0.5/r1 | 0.5/r5 | 0.5/r10 | 0.5/r100 | 0.7/r1 | 0.7/r10 | 0.7/r50 | 0.7/r100 |
> | ---------- | ------ | ------ | ------- | -------- | ------ | ------- | ------- | -------- |
> | HERO       | 0.24   | 1.34   | 1.75    | 3.83     | 0.17   | 0.77    | 1.08    | 2.28     |
> | XML        | 3.19   | 9.64   | 14.05   | 40.29    | 2.32   | 7.20    | 10.69   | 33.04    |
> | ReLoCLNet  | 3.74   | 11.01  | 15.62   | 40.29    | 1.92   | 6.75    | 9.84    | 31.47    |
> | CONQUER\*  | 5.48   | 15.45  | 22.33   | 51.63    | 3.66   | 10.12   | 15.87   | 42.64    |
> | SQuiDNet\* | 2.89   | 7.92   | 11.94   | 33.82    | 0.52   | 1.32    | 1.99    | 6.75     |
>
> * DiDeMo-FIG, VR
>
> | Method     | r1    | r5    | r10   | r100  |
> | ---------- | ----- | ----- | ----- | ----- |
> | HERO       | 8.48  | 26.73 | 39.52 | 84.46 |
> | XML        | 14.83 | 40.39 | 53.95 | 91.53 |
> | ReLoCLNet  | 14.08 | 37.18 | 50.88 | 91.30 |
> | CONQUER\*  | 14.83 | 40.39 | 53.95 | 91.53 |
> | SQuiDNet\* | 16.94 | 44.58 | 59.26 | 91.53 |
>
> Overall, CONQUER is the best-performing method. Although SQuiDNet achieves the best in VR because it continues video-level learning in the second stage, it shows *unstable* performance in VCMR. **We suggest not entangling video-level and moment-level learning** in the training stage for the fine-grained setting. Introducing more fine-grained information when learning video-level retrieval might interfere with moment localization.
>
> We will add this analysis to the paper later.

---

> ### Author Rebuttal · Authors · 2024-08-18
>
> > The tables take up a lot of space.
>
> Thanks! We will rearrange the paper's layout to add our new experiment analysis.
>
> **2. It would be nice to have a study on how close you are to being fully deterministic.**
>
> We have this part in the supplementary materials(`supplementary_materials.pdf`) in Table 2. We count the number of queries that have more than 1 potential positive video moment in both COG and FIG settings. We find the number drops a lot in our FIG setting compared to COG. It indicates that our FIG datasets are much more deterministic than previous ones. We will move it to the main paper PDF.
>
> **3. The performance of a model that would completely solve the benchmark is bounded by LLMs or engineered approaches.**
>
> Thanks for your advice. Our pipeline integrates some human prior knowledge for assistance. In our annotation pipeline, we use the previous human coarse annotations for better reference guidance. In our fine-tuning stage, we disturb the captions based on human annotations. We combine the advantages of both machine and human expertise, which helps mitigate the bound limitation. The annotation quality is acknowledged by human evaluators (the manual evaluation in Section B.2 of supplementary materials `supplementary_materials.pdf`), so the model that would completely solve the benchmark would also address real human demands, although the bound exists in nature. We will conduct more verification experiments in the future.
>
> **4. Is the goal then to obtain a completely end-to-end model that would have the capabilities of all these models combined? Do you have thoughts about that?**
>
> Thanks for your insightful suggestions. This goal would be the future direction, which would help to construct a more fast system for things we do. The "Dense Video Captioning" task aims to finish it in an end-to-end model, which localizes event moments from videos and generates captions for the moments. To obtain an end-to-end model that has the capabilities of all these models combined, the vanilla idea is to design a dense video captioning model and train it with our FIG data. The dense video captioning model can be developed based on existing video LLM architecture for they have generalization abilities. In our pipeline, we use prompts to control the LLM/LVM to focus on information that is important to humans. Thus furthermore, I think some intermediate pretraining tasks are also important to achieve the goal, like telling the differences when given two or more video clips because, in real life, not all the details are important, which would help the model to generate captions that better cover the crucial semantics of videos that make the video distinctive. This would be an interesting discussion and we may add it to the paper.

---

### Official Review · Reviewer_uTUV · 2024-07-25
**VERIFIED: A Video Corpus Moment Retrieval Benchmark for Fine-Grained Video Understanding**

**Rating:** 8
**Confidence:** 3
**Correctness:** Yes

**Review:**

This paper presents a valuable contribution to the field of fine-grained video understanding by proposing a VCMR (Video Clip Moment Retrieval) benchmark. The work is centered around a novel automatic annotation pipeline called VERIFIED, which leverages existing LLM/VLMs to enhance video clip annotations.

Pros:
1. Solid starting point
2. Carefully desgined annotating framework
3. Experiments show effectiveness

Cons:
1. Dependency on LLM/VLM quality
2. Limited evaluation scope

**Strengths:**

1. The starting point of the paper is solid. False negatives do exist in existing tasks and interfere with the performance of the models. A fine-grained VCMR benchmark will surely help the evalution and building of retrieval models.
2. The annotating framework VERIFIED is carefully designed and reasonable. The propose of Fine-Granularity Aware Noise Evaluator is especially interesting. It fully utilizes existing VLM models to produce accurate content, avoiding hallucinations. The implementation details are clear and thorough, making it easy to reimplement.
3. The user study and visualization demonstrates the effectiveness of fine-grained captioning.

**Additional Feedback:**

Nil

**Clarity:**

The paper is overall clearly written. But math notations used in method part are onerous and mostly unnecessary.

**Documentation:**

The authors give sufficient implementation details in 3.4 which seems adequate for reproducibility.

**Ethics:**

The benchmark is built on previous datasets. Still, the author may need to investigate further to ensure consent to use those data.

**Limitations:**

The authors mentioned the limitations in the conclusion, that more general video tasks are unexplored.

**Opportunities For Improvement:**

1. Despite the effort of the authors to avoid hallucinations, like any other automated generated data with LLMs, the quality of the benchmark is limited by the VLM/LLMs used in constructing. A small subset labeled by human for comparison will be nice.
2. The evaluation of benchmark includes serveral state-of-the-art methods, but the results are only tested on FIG. To my understanding, if the quality of the training set improves, the results on COG test set should also be improved since less noise is introduced during the training.
3. The math notations used in method part are onerous and mostly unnecessary. The writing will be much clear if the notations are simplified.

**Relation To Prior Work:**

Yes

**Summary And Contributions:**

This paper proposes a VCMR benchmark for fine-grained video understanding. It first defines a fine-grained VCMR setting, then proposes VERIFIED, an automatic fine-grained video clip annotation pipeline, inclduing Statics and Dynamics Enhanced Captioning and Fine-Granularity Aware Noise Evaluator with the help of existing LLM/VLMs. The benchmark is evaluated across several sota VCMR approaches with comprehensive experiments and analysis.

---

> ### Author Rebuttal · Authors · 2024-08-18
>
> **1. Dependency on LLM/LVM quality. A subset of human annotations for comparison will be nice.**
>
> Thanks for your suggestion! Following your suggestion, we construct a subset of human annotations, where we ask human annotators to integrate detailed content on static and dynamic information when writing captions. We find human captions exhibit a comparable level of annotation quality compared to those of the LLM/LVM pipeline, although they may focus on different aspects of the videos. We have already evaluated our fine-grained dataset through manual assessment (see details in Section B.2 of supplementary materials `supplementary_materials.pdf`), and the consistency is very high, showing that the quality of our annotations is recognized by humans. We will move this human check part to the main paper later.
>
> There are several examples of human annotations in this rebuttal PDF. We will release this subset in our dataset later.
>
> **2. Limited evaluation scope. The results are only tested on FIG. The results of the COG test set should also be improved since less noise is introduced during training.**
>
> Thanks for your suggestion! We not only have already conducted tests on FIG but also the COG setting. This can be found in the supplementary materials(`supplementary_materials.pdf`) in Table 4 (VCMR task). We will move this part to the main paper later.
>
> We test the models in both FIG and COG scenarios when the models are trained with FIG data. We find that the results in COG are similar to when the models are trained with COG data. The reasons might be:
>
> (i) The COG annotations can be seen as a subset of the FIG annotations because the FIG reserves the previous coarse information with additional detailed content. The FIG training equips the model with the ability to understand fine-grained information and distinguish similar moments. However, many queries in the COG test do not have fine-grained information, so for two similar video moments, they are both suitable for the coarse query, making such an ability not required in the COG setting.
>
> (ii) We select the annotation with the highest score to maintain the number of annotations the same as the previous COG Charades, DiDeMo, and ActivityNet Captions. However, the new annotations are filled with more detailed information and a larger vocabulary, making it harder for models to learn with the same training data scale. It's easy to get more annotations from our pipeline by tuning some hyperparameters and we will release a second version of our FIG datasets with more annotations later.
>
> **3. Math notations.**
>
> Thanks for your suggestion! We will simplify mathematical notations to make the writing more clear.

---

> > ### Comment · Reviewer_uTUV · 2024-09-02
> > **Response to Rebuttal**
> >
> > I appreciate the authors’ efforts in addressing my questions. While some reviewers have expressed concerns that the benchmark lacks methods for improving existing models, I believe that the primary purpose of a benchmark is to establish reasonable tasks that serve as indicators of a model’s capabilities. Enhancing existing models goes beyond the scope of what benchmarks are intended for. Therefore, I would like to keep my rating.

---

### Decision · Program_Chairs · 2024-09-26

**Decision:**

Accept (Poster)

**Comment:**

Of the four reviews, three reviews gave clear acceptance with one review reject. However, the authors gave extensive rebuttal and clarification to the reject reviewer's questions and comments, which mostly addressed the concerns of this reviewer. But, the reviewer didn't respond. Overall, this paper addresses a difficult problem with increasing importance that can benefit the wider community, and it's timely.